# *Plasmodium* RON11 triggers biogenesis of the merozoite rhoptry pair and is essential for erythrocyte invasion

David Anaguano[1,2], Opeoluwa Adewale-Fasoro[3,4], Grace W. Vick[2,5], Sean Yanik[3,4], James Blauwkamp[6], Manuel A. Fierro[1,2¤], Sabrina Absalon[6], Prakash Srinivasan[3,4], Vasant Muralidharan[1,2]*

1 Department of Cellular Biology, University of Georgia, Athens, Georgia, United States of America, 2 Center for Tropical and Emerging Global Diseases, University of Georgia, Athens, Georgia, United States of America, 3 Department of Molecular Microbiology and Immunology, and Johns Hopkins Malaria Research Institute, Johns Hopkins Bloomberg School of Public Health, Baltimore, Maryland, United States of America, 4 The Johns Hopkins Malaria Research Institute, Baltimore, Maryland, United States of America, 5 Department of Infectious Diseases, College of Veterinary Medicine, University of Georgia, Athens, Georgia, United States of America, 6 Department of Pharmacology and Toxicology, Indiana University School of Medicine, Indianapolis, Indiana, United States of America

¤ Current address: Department of Biomedical Sciences, College of Veterinary Medicine, Iowa State University, Ames, Iowa, United States of America
* vasant@uga.edu

**Data Availability Statement:** All relevant data are within the paper and its Supporting Information files.

## Abstract

Malaria is a global and deadly human disease caused by the apicomplexan parasites of the genus *Plasmodium*. Parasite proliferation within human red blood cells (RBCs) is associated with the clinical manifestations of the disease. This asexual expansion within human RBCs begins with the invasion of RBCs by *P. falciparum*, which is mediated by the secretion of effectors from 2 specialized club-shaped secretory organelles in merozoite-stage parasites known as rhoptries. We investigated the function of the Rhoptry Neck Protein 11 (RON11), which contains 7 transmembrane domains and calcium-binding EF-hand domains. We generated conditional mutants of the *P. falciparum* RON11. Knockdown of RON11 inhibits parasite growth by preventing merozoite invasion. The loss of RON11 did not lead to any defects in processing of rhoptry proteins but instead led to a decrease in the amount of rhoptry proteins. We utilized ultrastructure expansion microscopy (U-ExM) to determine the effect of RON11 knockdown on rhoptry biogenesis. Surprisingly, in the absence of RON11, fully developed merozoites had only 1 rhoptry each. The single rhoptry in RON11-deficient merozoites were morphologically typical with a bulb and a neck oriented into the apical polar ring. Moreover, rhoptry proteins are trafficked accurately to the single rhoptry in RON11-deficient parasites. These data show that in the absence of RON11, the first rhoptry is generated during schizogony but upon the start of cytokinesis, the second rhoptry never forms. Interestingly, these single-rhoptry merozoites were able to attach to host RBCs but are unable to invade RBCs. Instead, RON11-deficient merozoites continue to engage with RBC for prolonged periods eventually resulting in echinocytosis, a result of secreting the contents from the single rhoptry

**Funding:** NIH/NIAID R56 AI173133 (V.M.), T32 AI060546 (G.W.V),T32 AI138953 (S.Y.), Office of the Vice-President for Research at UGA (D.A.), the CTEGD TIPS Fellowship and Daniel G. Colley Training in Parasitology Fund (D.A.), Johns Hopkins Malaria Research predoctoral fellowship (O.A-F), and in part by the Johns Hopkins MalariaResearch Institute and the Bloomberg Philanthropies (P.S.). The funders had no role in study design, data collection and analysis, decision to publish, or preparation of the manuscript.

**Competing interests:** The authors have declared that no competing interests exist.

**Abbreviations:** APS, ammonium persulfate; CP, centriolar plaque; HA, hemagglutinin; hpi, hours post-invasion; IMAC, immobilized-metal affinity chromatography; NHS, N-hydroxysuccinimide; PBS, phosphate-buffered saline; PCC, Pearson's correlation coefficient; PFA, paraformaldehyde; PMIX, Plasmepsin IX; PV, parasitophorous vacuole; RBC, red blood cell; SDS, sodium dodecyl sulfate; U-ExM, ultrastructure expansion microscopy.

into the RBC. Together, our data show that RON11 triggers the de novo biogenesis of the second rhoptry and functions in RBC invasion.

## Introduction

Malaria remains a devastating disease that affects approximately 249 million people worldwide in 2022, with the African region accounting for over 95% of the cases [1]. The main causative agent of malaria is the unicellular protozoan parasite *Plasmodium falciparum*, which belongs to the apicomplexan phylum, along with other medically important organisms such as *Toxoplasma gondii* and *Cryptosporidium parvum*. Clinical symptoms of malaria in humans are associated with the continuous cycle of invasion of red blood cells (RBCs), development and replication within RBCs, and egress from RBCs to release new merozoites that perpetuate the cycle. Invasion is a process that has been previously studied as a promising target for the development of vaccination and antibody treatment strategies [2,3], which underscores the importance of understanding this process for the development of clinical interventions.

Invasion is a rapid and complex process involving multiple steps. Newly egressed merozoites initiate an early attachment, leading to RBC deformation and merozoite reorientation [4–6]. Once its apical end is reoriented towards the RBC, the merozoite appears to form a pore in the RBC membrane, secretes proteins from rhoptries into the RBC, which together with micronemal proteins on the merozoite membrane, form an irreversible tight junction at the interface merozoite-RBC [6–11]. Subsequently, the merozoite utilizes its actinomyosin motor to push through the tight junction and internalize within the newly formed parasitophorous vacuole (PV) [7,9,12–15]. Within the PV, merozoites develop into the ring stage, progressing into the metabolically active forms, trophozoites, and finally into the replicative form, schizonts, which make 16 to 32 daughter merozoites. The parasite then initiates a cascade of events leading to the rupture of the RBC and PV membranes, resulting in the egress of merozoites [16].

Rhoptries are specialized club-shaped secretory organelles in apicomplexans known to be essential for parasite proliferation. Even though rhoptries have conserved roles among apicomplexans, their numbers vary between species, for instance, *Cryptosporidium* spp. have a single rhoptry, *Plasmodium* spp. have 2 in the asexual stages, and *Toxoplasma* spp. have 8 to 12 rhoptries [8,17–19]. In *P. falciparum*, rhoptries are located at the apical end of the merozoite and secrete proteins that facilitate different steps during invasion. Rhoptries are internally compartmentalized into 2 regions: the neck and bulb, storing specific groups of proteins that are secreted at each step of invasion [8,20,21]. Rhoptry neck proteins are assumed to be released first, primarily mediating attachment to RBCs and the formation of the tight junction through secretion of proteins from the RON complex [6,22]. In contrast, rhoptry bulb proteins are secreted after tight junction formation and are associated with proper development within the PV, such as RAP1 and RAP2, which form the RAP complex [10,23]. During invasion, rhoptries fuse their necks and bulbs to form a single rhoptry [24]. After invasion is completed, this single rhoptry is assumed to be disassembled, as rhoptries are absent during early ring stages. During schizogony, rhoptries are then formed de novo by the fusion of Golgi-derived vesicles [25,26].

Rhoptry biogenesis begins early during schizogony, where the rhoptries remain associated with the cytoplasmic extensions of the outer centriolar plaque (CP) branches in a 1:1 ratio with each new rhoptry forming de novo through each remaining mitotic event, until the last event where this ratio is broken and each outer CP branch is observed with 2 rhoptries [25,27,28].

The final rhoptry pairs undergo de novo biogenesis instead of dividing from a common rhoptry precursor [27], as was previously assumed. Rhoptries maintain a bulb shape throughout most of their biogenesis process, adopting their final shape only after merozoite segmentation, with the formation of the neck in each rhoptry [25,27]. However, the molecular factors regulating rhoptry biogenesis, their structure, and internal compartmentalization are unknown.

Rhoptry neck protein 11, or RON11 (PF3D7_1463900), is a seven-transmembrane domain protein with a pair of calcium-binding EF-hand domains at its C-terminus that is conserved in Apicomplexans. Initial attempts to disrupt *P. berghei* RON11 were unsuccessful [29,30], while a recent knockdown showed a potential role in sporozoite invasion and gliding motility [31]. In contrast, RON11 disruption in *T. gondii* showed only a minor fitness defect, suggesting a non-essential role [32]. However, nothing is known about the role of RON11 in *P. falciparum*. In this study, we employed the tetR-DOZI conditional knockdown system [33] to investigate the function of RON11 in the *P. falciparum* intraerythrocytic asexual life cycle. Our findings revealed that RON11 is essential for the intraerythrocytic growth of *P. falciparum*. Parasites lacking RON11 are unable to invade RBCs. Anti-RON11 antibodies are able to block merozoite invasion into RBCs suggesting that RON11 plays an essential role during invasion. Surprisingly, RON11 knockdown results in merozoites that have a single rhoptry. These single rhoptry merozoites are unable to complete invasion, even though they attach and secrete the contents from the single rhoptry into RBCs. Collectively, our data show that RON11 plays an essential role in regulating the de novo formation of the second rhoptry during merozoite segmentation and required for merozoite invasion.

## Results

### RON11 is essential for *P. falciparum* intraerythrocytic life cycle

To investigate the role of RON11 in *P. falciparum*, we generated RON11 conditional knockdown parasite lines employing the tetR-DOZI aptamer system [33], termed as RON11[apt]. Using CRISPR/Cas9 technology, we integrated the tetR-DOZI system into the endogenous loci of RON11, leading to the expression of a hemagglutinin (HA)-tagged protein under the regulation of anhydrotetracycline (aTc) (Fig 1A). PCR analysis of genomic DNA from RON11[apt] parasites showed correct integration at the endogenous loci (Fig 1B). To test the efficiency of the knockdown system, we evaluated protein expression both in the presence (+-RON11) and absence (-RON11) of aTc via western blotting. Lysates from synchronized RON11[apt] ring stage parasites cultured without aTc for 44 h post-invasion (hpi) (until schizont stage) exhibited around 97% of reduction in protein expression (Fig 1C), confirming the system's suitability for studying RON11 function. Parasite growth was measured using flow cytometry (Fig 1D). These data show that RON11 is essential for parasite expansion, as -RON11 parasites failed to progress past their first life cycle (Fig 1D).

Furthermore, we assessed the subcellular localization of RON11, which was previously observed in rhoptries of *P. berghei* schizonts [31] (Fig 1E). RON11 was localized with 2 well-known markers of the rhoptry neck (RON4) and rhoptry bulb (RAP1) (Fig 1E). Our data showed the presence of RON11 in the rhoptry neck, as suggested by its colocalization with RON4 (Fig 1E, top panels, and 1F). RON11 did not colocalize with RAP1, suggesting that RON11 is confined to the neck region (Fig 1E, bottom panels, and 1F). Additionally, we observed RON11 at the parasite periphery of newly invaded ring-stage parasites (S1 Fig), suggesting that RON11 might localize to the PV after invasion, similar to RAP1 [10,27,34]. Taken together, these data show RON11 localizes to the rhoptry neck in schizonts, and in early-ring-stage parasites, it localizes at the parasite periphery.

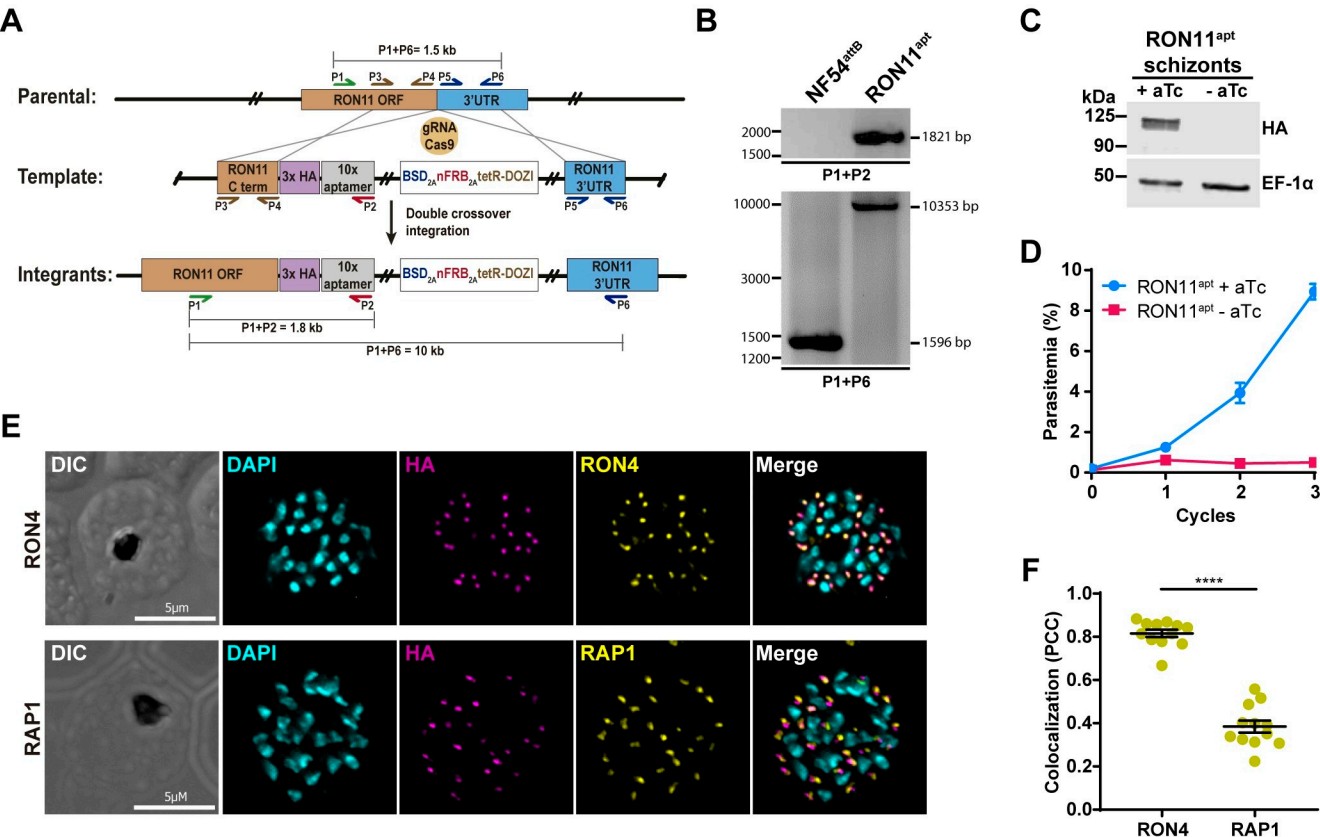

**Fig 1. RON11 is essential for intraerythrocytic growth.** **(A)** Schematic showing the integration of the repair plasmid that introduces the tetR-DOZI system in the genomic loci of RON11. **(B)** PCR confirming integration at the RON11 locus. Amplicons were amplified from genomic DNA isolated from the mutant and the parental line (NF54[attB]) [62]. Primers were designed to amplify 2 regions as shown. **(C)** Western blot of lysates from RON11[apt] schizont grown in the presence or absence of 0.5 μm aTc. Lysates were probed with antibodies against HA (RON11) and EF1α (loading control). The protein marker sizes are shown on the left ($n = 4$ biological replicates). **(D)** Growth of RON11[apt] parasites in the presence or absence of 0.5 μm aTc. Synchronous parasites were collected every 48 h, stained, and measured via flow cytometry. One representative data set of 3 biological replicates shown ($n = 3$ technical replicates; error bars = standard deviation (SD), the underlying data can be found in S1 Data). **(E)** IFAs showing the localization of RON11 in RON11[apt] schizont with respect to 2 rhoptry markers: RON4 (neck) and RAP1 (bulb). Images from left to right are phase-contrast, DAPI (nucleus, cyan), anti-HA (magenta), anti-RON4, or RAP1 (yellow), and fluorescence merge. Z stack images were deconvolved and projected as a combined single image. Representative images of 3 biological replicates. **(F)** Quantification of the colocalization of RON11[apt] with RON4 and RAP1 as determined by the PCC ($n = 3$ biological replicates, 4 schizonts per replicate, the underlying data can be found in S1 Data). Error bars = SEM; ****$p < 0.0001$ by unpaired two-tailed $t$ test. HA, hemagglutinin; IFA, immunofluorescence assay; IgG, immunoglobulin G; MBP, maltose binding protein; PCC, Pearson's correlation coefficient; SD, standard deviation; SEM, standard error of the mean.

## RON11 is required for merozoite invasion

RON11 is known to be highly expressed in schizont stages [35,36], and we have shown its knockdown inhibits parasite growth (Fig 1D). Therefore, we wanted to determine if RON11 is necessary for schizont development, egress or merozoite invasion. To define the specific stage of the intraerythrocytic lifecycle that requires the function of RON11, we monitored the progression of synchronized RON11[apt] parasites through the asexual life cycle, both in the presence and absence of aTc. Samples were collected at all the stages, starting with tightly synchronized schizonts (46–48 hpi), observed using Hema3-stained thin blood smears and light microscopy. Our findings show that, both +RON11 and -RON11 parasites developed normally from rings through schizonts, and after completion of schizogony, daughter merozoites egressed at 48 hpi (Fig 2A). However, upon RON11 knockdown, the egressed merozoites did not produce rings, instead, they accumulated around the RBCs (Fig 2A). To quantify our

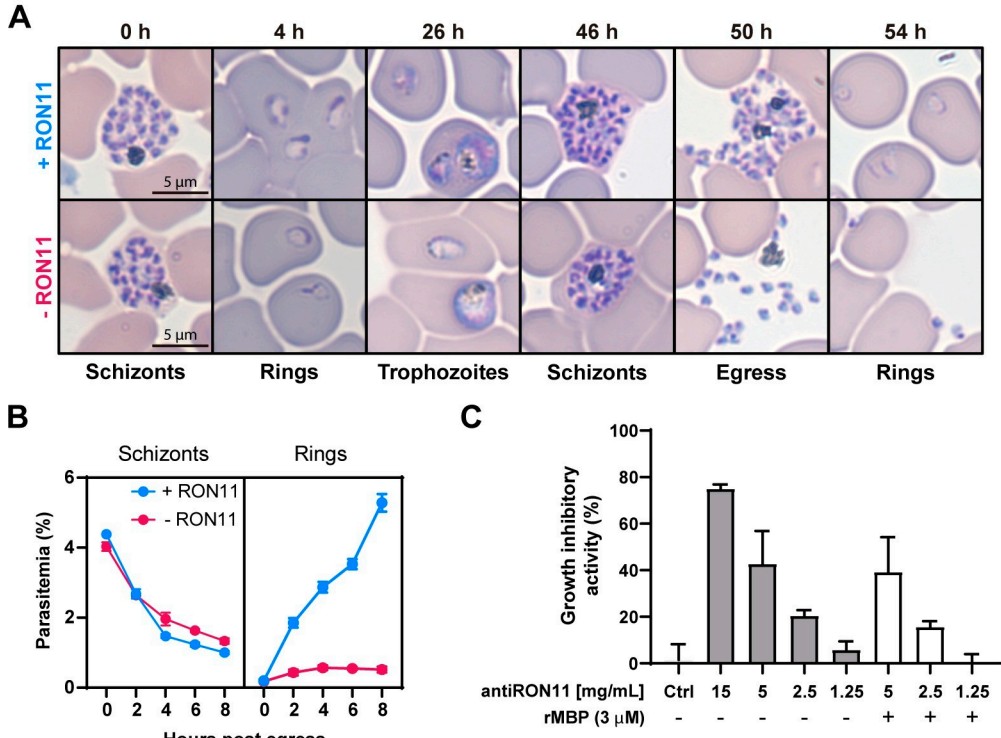

**Fig 2. RON11 is required for merozoite invasion. (A)** Representative Hema-3-stained blood-smears showing the development of RON11$^{apt}$ parasites in the presence or absence of 0.5 μm aTc. Synchronous parasites were smeared, stained, and imaged by light microscopy. **(B)** Development of synchronous RON11$^{apt}$ parasites over 8 h post-egress in the presence or absence of 0.5 μm aTc. Schizonts were synchronized using 2 μm Compound 1 and collected every 2 h, stained, and measured via flow cytometry to distinguish between rings and schizonts. One representative data set of 3 biological replicates shown (*n* = 3 technical replicates; error bars = SD; the underlying data can be found in S1 Data). **(C)** In vitro neutralization assay was performed against 3D7 parasites using total IgG purified from rats immunized with rMBP-RON11 (N-term) or non-immune rat IgG (8 mg/ml) as the negative control (labeled Ctrl). Potential influence of anti-MBP antibodies was evaluated by pre-incubating IgG with rMBP (3 μm). Parasites were stained with SYBR green and parasitemia was measured by flow cytometry. Growth inhibitory activity is represented as the percentage (%) of inhibition relative to the "no-IgG" control (*n* = 2–4 biological replicates in duplicate; error bars = SEM; the underlying data can be found in S1 Data).

microscopy-based observations, we used flow cytometry to measure parasite transition from schizonts into rings. We confirmed that even though schizonts egressed in the absence of RON11, rings were not detected at any time point post-egress (Fig 2B). Together these findings show that the knockdown of RON11 prevents merozoite invasion, while having no effect on intraerythrocytic development or on the egress of daughter merozoites.

We hypothesized that RON11 may have a specific function in merozoite invasion. To independently test this hypothesis, we generated antibodies against the N-terminus of RON11 (aa32-442) (S2A and S2B Fig). If anti-RON11 antibodies could block parasite invasion into the host RBC, then this would suggest that RON11 plays a specific role in merozoite invasion. Western blot analysis detected a single high molecular weight protein of the expected size in schizont lysate (S2D Fig), showing that anti-RON11 antibodies were specific for RON11. We also confirmed the specificity of the anti-RON11 antibodies using immunofluorescence microscopy (S2E Fig). The anti-RON11 antibody staining was colocalized with the anti-HA staining in the RON11$^{apt}$ (S2E Fig). We then evaluated the ability of anti-RON11 antibodies to block merozoite invasion and therefore reduce parasite multiplication. In these neutralization

assays, parasites were incubated with anti-RON11 antibodies and their ability to proliferate was measured. *P. falciparum* parasites (3D7 strain) when incubated with various concentrations of total anti-RON11 IgG. Parasite growth was measured by flow cytometry (Fig 2C). Antibodies against RON11 exhibited concentration-dependent growth inhibitory activity, with around 80% growth inhibition at a concentration of 15 mg/ml (Fig 2C). This suggests that the N-terminus of RON11 is accessible to antibodies during merozoite invasion and may interfere with its function in RBC invasion. Together with our RON11 conditional knockdown data, these findings support a putative role for RON11 in merozoite invasion.

## RON11 is not required for merozoite attachment or eventual rhoptry discharge

Merozoite invasion is a complex multistep process, which involves the participation of multiple regulatory factors. In general terms, invasion can be divided into several discrete steps: attachment, apical end reorientation, RBC deformation, and pore formation, rhoptry secretion and tight junction formation, internalization and PVM sealing, and echinocytosis [4,37]. To dissect the specific role of RON11 during merozoite invasion, we investigated the following specific events during invasion: attachment, deformation, rhoptry secretion, and echinocytosis.

First, we used live-cell imaging to examine the merozoite-RBC interactions in recently egressed RON11$^{apt}$ merozoites in the presence and absence of RON11 (Figs 3A and S3 and S1 and S2 Movies). As shown previously (Fig 2A and 2B), RON11 knockdown completely inhibited invasion in merozoites that induced deformation ($n = 0/53$), in contrast to the +RON11 merozoites in which 77.2% ($n = 27/35$) of merozoites invaded successfully (Fig 3A and 3B). These invasion defects in -RON11 merozoites were observable even though these merozoites were capable of generating strong deformations of the RBC membrane (Figs 3B, 3C and S3D) similar to +RON11 merozoites ($p = 0.1122$). The -RON11 merozoites were capable of inducing echinocytosis ($n = 33/44$) in the attached RBCs (Fig 3D and S2 Movie). We also noticed that -RON11 merozoites continue to engage with the RBCs (deformation) for longer periods of time and remain attached to RBCs, eventually leading to echinocytosis (Figs 3C, 3D and S3C). However, no significant difference between +RON11 and -RON11 merozoites was observed in the time that it took the merozoites from their initial contact with RBCs to the beginning of echinocytosis (Fig 3D). Altogether, these data suggest that RON11 is not required for initial attachment, and deformation of RBCs, but is essential for internalization into the host cell. The longer deformation eventually leads to echinocytosis, which is thought to be indicative of secretion of contents from rhoptries into the RBC [6]. Our data suggests that while -RON11 parasites are capable of inducing echinocytosis though the timing of echinocytosis is prolonged in the absence of RON11 (Figs 3D, 3E, S3B and S3C).

To confirm our observation that there was no defect in the attachment of -RON11 parasites to host RBCs, we used a cytochalasin-D-based strategy to observe merozoite-RBC interactions [10]. Cytochalasin D is an actin polymerization inhibitor that disrupts the activity of the merozoite actomyosin motor, preventing internalization but not affecting the previous steps on merozoite invasion, such as attachment and rhoptry secretion [9,10,38]. Samples for analysis were collected from Compound 1 treated mature schizonts that were washed and allowed to invade fresh RBCs in the presence of cytochalasin D (S3E Fig).

We used light microscopy to quantify +RON11 and -RON11 merozoites attached to RBCs. In merozoites incubated in cytochalasin D, no significant difference in the numbers of attached +RON11 or -RON11 merozoites was observed (S3F Fig). When merozoites were allowed to invade without cytochalasin D, we found more attached -RON11 merozoites,

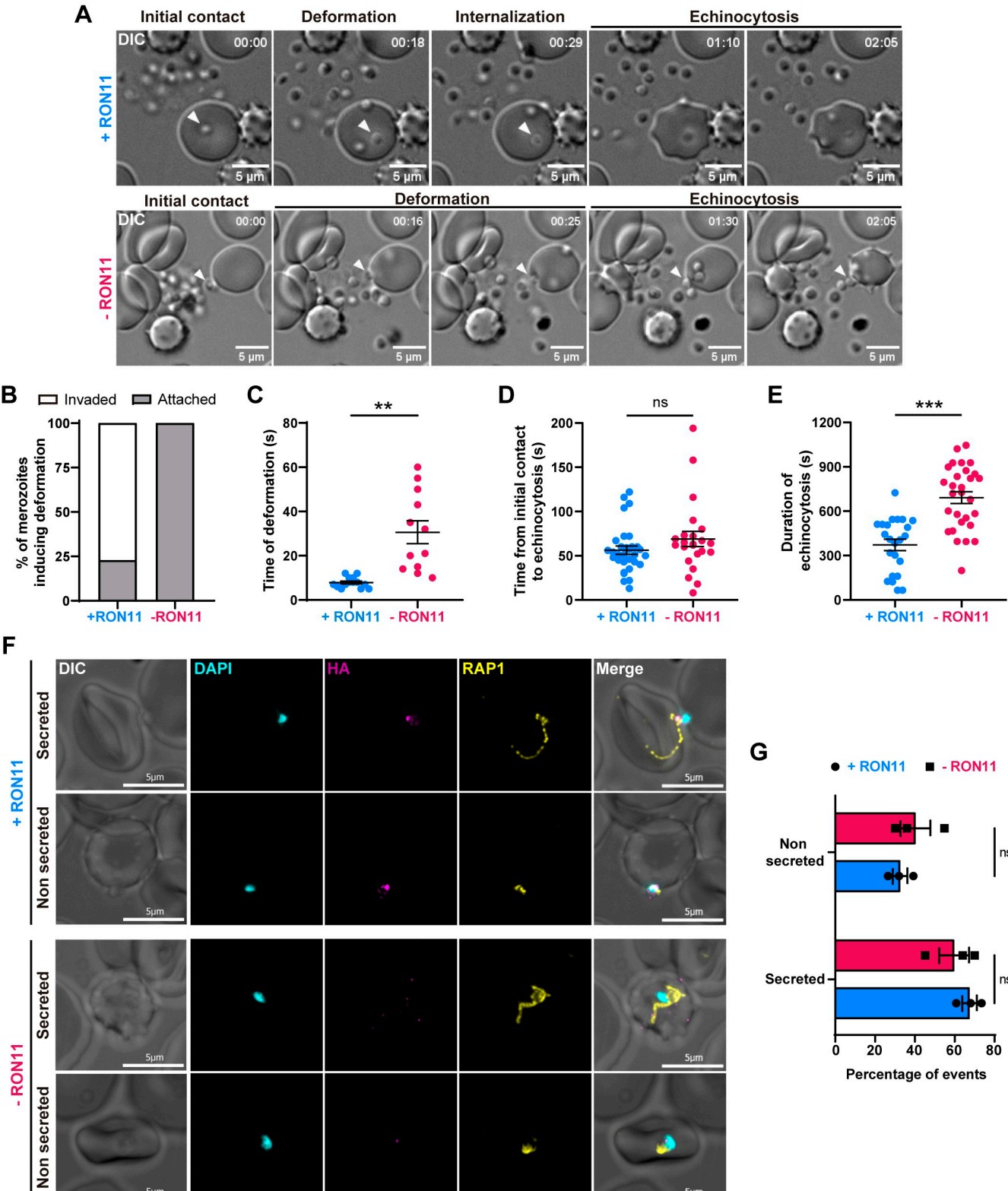

**Fig 3. RON11 is not required for merozoite attachment nor rhoptry secretion. (A)** Representative time-course images from S1 Movie (+RON11) and S2 Movie (-RON11) showing RON11^apt merozoites (arrowhead) interacting with RBCs in the presence or absence of 0.5 μm aTc. One representative data set of 3 biological replicates shown. **(B)** Quantification of merozoites that induced deformation in RBCs that completed invasion or remained attached (*n* = 3

biological replicates, 35 merozoites for +RON11, and 53 for -RON11; the underlying data can be found in S1 Data). **(C)** Duration of merozoite-induced RBC deformation quantified using live video microscopy ($n = 3$ biological replicates, 14 merozoites for +RON11 and 12 for -RON11; error bars = SEM; $^{**}p < 0.01$ by unpaired two-tailed *t* test; the underlying data can be found in S1 Data). **(D)** Duration between first contact of merozoites with RBCs and echinocytosis quantified using live video microscopy ($n = 3$ biological replicates, 35 merozoites for +RON11 and 53 for -RON11; error bars = SEM; ns = non-significant by unpaired two-tailed *t* test). **(E)** Duration of echinocytosis quantified using live video microscopy ($n = 3$ biological replicates, 23 merozoites for +RON11 and 29 for -RON11; error bars = SEM; $^{***}p < 0.001$ by unpaired two-tailed *t* test; the underlying data can be found in S1 Data). **(F)** Representative IFAs showing RON11[apt] attached-merozoites secreting RAP1 into RBCs in the presence or absence of aTc, after incubation with cytochalasin D. Images from left to right are phase-contrast, DAPI (nucleus, cyan), anti-HA (magenta), anti-RAP1 (yellow), and fluorescence merge. Z stack images were deconvolved and projected as a combined single image. Representative images of 3 biological replicates. **(G)** Quantification of RON11[apt] attached-merozoites secreting RAP1 into RBCs. Slides were blinded and secretion events were scored and represented as the percentage of events per 50 attached merozoites ($n = 3$ biological replicates, 50 events per replicate. Error bars = SEM; ns = non-significant by unpaired two-tailed *t* test; the underlying data can be found in S1 Data). RBC, red blood cell.

compared to the cytochalasin-D-treated +RON11 merozoites (S3G Fig). As expected, the number of attached merozoites dropped significantly when cytochalasin D was removed from +RON11 merozoites, as they were able to invade RBCs and form rings (S3G Fig). Together, these data demonstrated that RON11 is not required for merozoite attachment to RBCs.

Our live imaging data suggests that echinocytosis, albeit delayed, occurs in -RON11 parasites (Fig 3A, 3D and 3E). To confirm that -RON11 parasites are able to secrete rhoptry contents into the host RBC, we used IFAs to analyze the secretion of rhoptry proteins from cytochalasin-D-treated merozoites into RBCs in both +RON11 and -RON11 parasites. We selected RAP1 as a rhoptry secretion marker because it has been previously shown to be secreted into RBCs cytoplasm in the presence of cytochalasin D [10]. We found that -RON11 merozoites were able to secrete RAP1 into RBCs similarly to +RON11 merozoites (Fig 3F). Comparing the number of attached merozoites that secreted RAP1 into RBCs versus the ones who did not, no significant differences were observed between +RON11 and -RON11 parasites (Fig 3G). These results, together with our observation of prolonged deformation prior to echinocytosis by live video microscopy, suggest that RON11 deficient merozoites are able to eventually secrete rhoptry contents into the RBC.

## RON11 knockdown results in merozoites with one rhoptry

Given previous observations that other rhoptry membrane-associated proteins, such as RAMA, play a role in rhoptry formation and transport of rhoptry proteins [39], we sought to determine whether RON11 has a similar function in rhoptry biogenesis. Therefore, to determine if RON11 functions in rhoptry biogenesis, we employed ultrastructural expansion microscopy (U-ExM) in combination with the general protein stain N-hydroxysuccinimide (NHS) ester. NHS-ester staining accumulates in regions of high protein density, which facilitates the observation of rhoptry structures in high resolution [27]. We analyzed tightly synchronized ML10-treated schizonts that were cultured in the presence or absence of aTc. ML10 is a highly specific inhibitor of the cGMP-dependent protein kinase (PKG) that arrests parasites prior to egress [40], allowing for a tighter synchronization.

We observed fully formed rhoptries exhibiting their distinctive structures: a slender and elongated neck at the apex, connected to an apical ring, and a distal rotund bulb (Fig 4A, top panels), as previously reported [27]. Strikingly, -RON11 merozoites had only 1 rhoptry (Fig 4A and 4B). These single rhoptries displayed no discernible structural alterations, as the neck, and the bulb, were still present in the proper orientation positioned at the apical end of the merozoite, as defined by the observed apical ring (Fig 4A, bottom panels). In rare cases, we observed -RON11 schizonts with several rhoptries amassed in regions that appeared to be located outside of merozoites (S4A Fig). Basal complexes were visible at their maximum contraction at the basal end of the analyzed merozoites [27], suggesting that RON11 knockdown had no impact on merozoite segmentation, and confirming that the analyzed merozoites were

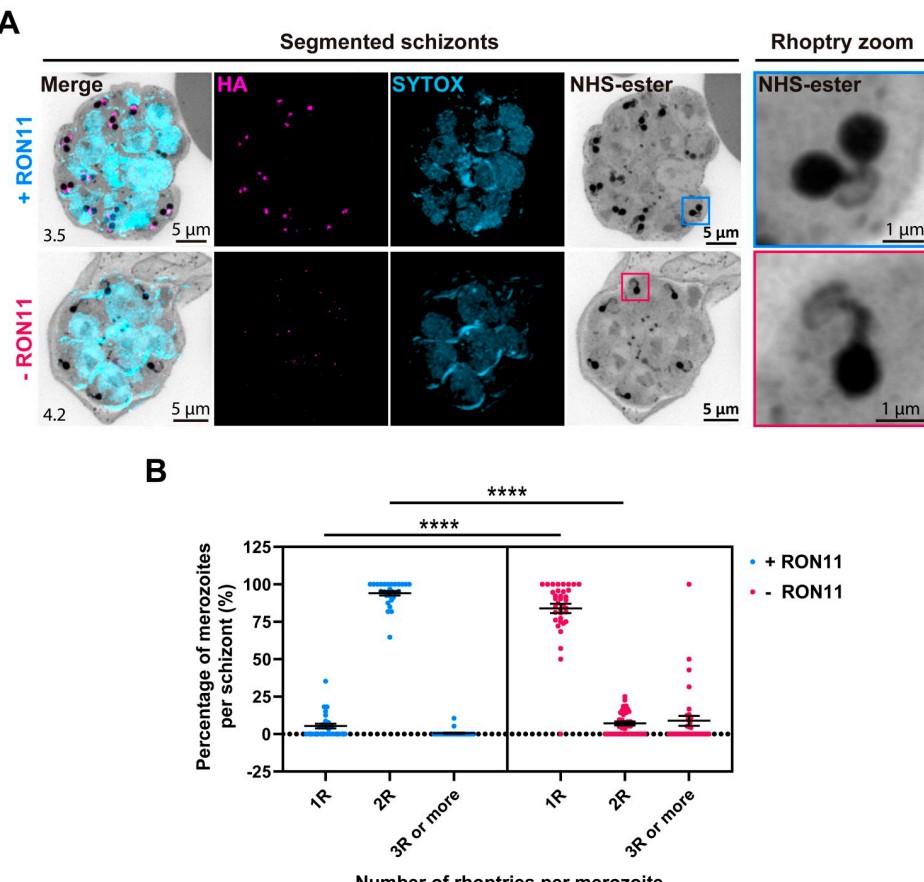

**Fig 4. RON11 knockdown generates merozoites with single rhoptries. (A)** Representative U-ExM images of RON11^apt schizonts showing the structure of rhoptries within fully developed merozoites in the presence or absence of 0.5 μm aTc. ML10-arrested schizonts were stained with NHS-Ester (grayscale), anti-HA (magenta), and the DNA dye SYTOX (cyan). Selected Z stack images were projected as a combined single image. Number on image = Z-axis thickness of projection in μm. **(B)** Quantification of the percentage of merozoites per schizont with single, dual, or multiple rhoptries in the presence or absence of 0.5 μm aTc. Samples were blinded and the number of rhoptries per merozoite within each schizont were counted. Merozoites were scored based on the number of nuclei observed ($n = 4$ biological replicates, 28 schizonts for +RON11 and 36 for -RON11; error bars = SEM; ****$p < 0.0001$ by unpaired two-tailed $t$ test; the underlying data can be found in S1 Data). HA, hemagglutinin; NHS, N-hydroxysuccinimide; U-ExM, ultrastructure expansion microscopy.

fully developed (Figs 4A and S4A). To validate this finding, we quantified the number of merozoites per schizont and found that the absence of RON11 had no effect on the total number of merozoites, even though the number of rhoptries was reduced by half (S4B and S4C Fig). To our knowledge, this marks the first observation of such a unique phenotype, single rhoptry merozoites (or changes in rhoptry numbers), in apicomplexan parasites.

To determine if RON11 knockdown disrupts the localization of rhoptry proteins, we analyzed the localization of other subcellular markers in segmented schizonts using U-ExM. We confirmed that RON11 localizes to the neck of fully formed rhoptries, along with its colocalization with RON4 (Fig 5A). These U-ExM data show that the localization of RON4 and RAP1 remained unaltered in the single rhoptries in -RON11 parasites (Fig 5A). Similarly, RON4 and RAP1 retained their localization in rhoptries of free -RON11 merozoites (Fig 5A).

Since -RON11 parasites had only 1 rhoptry, this suggests that there is a defect in the biogenesis of the second rhoptry. This raised the question, does single rhoptry in -RON11 schizonts

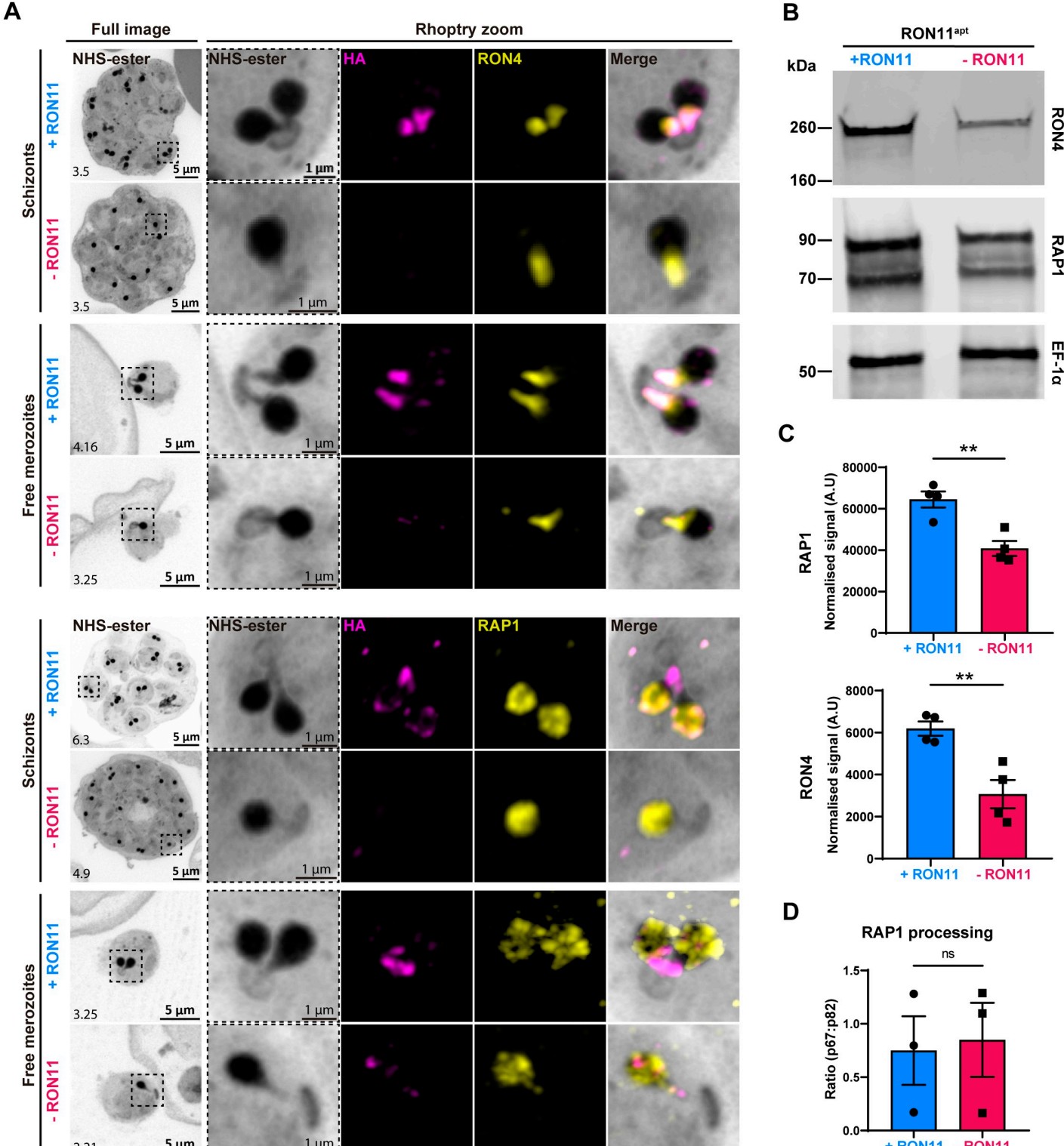

**Fig 5. RON11 knockdown does not affect the localization and processing of neck and bulb rhoptry proteins. (A)** Representative U-ExM images of RON11[apt] free merozoites and E64-arrested schizonts showing the localization of RON11, RON4, and RAP1 in the presence or absence of aTc. Parasites were stained with NHS-Ester (grayscale), anti-HA (magenta), and RON4 or RAP1 (yellow). Selected Z stack images were projected as a combined single image. Number on image = Z-axis thickness of projection in μm. **(B)** Western blot of parasite lysates isolated from E64-arrested RON11[apt] parasites in the presence or absence of aTc. Samples were probed with antibodies against RON4, RAP1, and EF1α (loading control). The protein marker sizes are shown on the left. Representative blot of 4 biological replicates shown. **(C)**

Quantification of RON4 and RAP1 in E64-arrested RON11$^{apt}$ parasites in the presence or absence of aTc. Band intensities were normalized to the loading control, EF1$\alpha$, and are presented as normalized arbitrary units (AU) ($n$ = 4 biological replicates; error bars = SEM; **$p$ < 0.01 by unpaired two-tailed $t$ test; the underlying data can be found in S1 Data). **(D)** Quantification of RAP1 processing in E64-arrested RON11$^{apt}$ parasites in the presence or absence of aTc. Band intensities were normalized to the ratio of processed RAP1 (p67/p82) ($n$ = 3 biological replicates; error bars = SEM; ns = non-significant by unpaired two-tailed $t$ test; the underlying data can be found in S1 Data). HA, hemagglutinin; NHS, N-hydroxysuccinimide; U-ExM, ultrastructure expansion microscopy.

have the same amount of rhoptry proteins as the 2 rhoptries in +RON11 parasites? To answer this question, we assessed the expression of RON4 and RAP1 in fully matured +RON11 and -RON11 schizonts (48 hpi) using western blotting. The total amount of RON4 and RAP1 was reduced by approximately half in -RON11 parasites (Fig 5B and 5C). This finding could be attributed to the reduction of the number of rhoptries by half, aligning with recent findings suggesting that rhoptry pairs form de novo during schizogony rather than through duplication from a single mother rhoptry [27].

Our data show that the single rhoptry -RON11 merozoites are unable to invade host RBCs. The biogenesis as well as the rhoptry biogenesis defect could be due to defective maturation of other rhoptry proteins by the rhoptry-localized aspartic protease, Plasmepsin IX (PMIX) [41–43]. PMIX processes several rhoptry proteins, including invasion ligands [41–43]. Therefore, we tested if RAP1 processing by PMIX is inhibited in -RON11 parasites (Fig 5D). We assessed the cleavage of RAP1 by PMIX into 67 kDa (p67) and 82 kDa (p82) forms in lysates collected from mature +RON11 and -RON11 schizonts (48 hpi) using western blotting (Fig 5D). The ratio between the cleaved forms of RAP1 (p67:p82) did not differ between +RON11 and -RON11 parasites. These data suggest that PMIX activity was intact in the single rhoptry -RON11 merozoites.

To rule out a generalized defect in the secretory pathway caused by the loss of RON11, we evaluated the expression of markers of other subcellular compartments, such as the micronemes (AMA1) and merozoite surface (MSP1). We used IFAs and western blotting on fully mature E64-treated schizonts cultured in the absence of RON11. We observed no effect on protein content or secretion of MSP1 to the merozoite surface under RON11 knockdown (S5A and S5B Fig). Similarly, AMA1 retained its micronemal and secreted locations in the absence of RON11 (S5C Fig). These data show that the loss of RON11 did not lead to a generalized defect in protein secretion.

## RON11 triggers the biogenesis of the last rhoptry pair

Rhoptries are first observed early during schizogony [27,44] and our data shows that -RON11 merozoites have a single rhoptry (Fig 4). When does rhoptry biogenesis fail during schizogony and how does this generate single rhoptry merozoites? Because schizogony is known to last approximately 15 h [45], we designed a rescue assay to define the window of time within schizogony where restoring RON11 might restore parasite growth during the final hours of the 48 h asexual life cycle. Synchronized ring-stage RON11$^{apt}$ parasites were grown with or without aTc and when these parasites reached schizogony (44 to 50 hpi), aTc was added back to the parasites at different time points (Fig 6). The ability of these schizonts to form rings was measured using flow cytometry. Supplementation with aTc as late as 46 hpi restored the ability of -RON11 schizonts to generate ring stage parasites (Fig 6). However, aTc supplementation after 46 hpi was unable to rescue growth (Fig 6). These findings suggested that RON11 has an essential role during the final hours of schizont development (Fig 6).

The ability of aTc to restore parasite growth as late as 46 hpi was surprising and suggested that the biogenesis of rhoptries can occur late during schizogony. To test if aTc addition during schizogony results in merozoites with 2 rhoptries, we added aTc to -RON11 parasites at 46

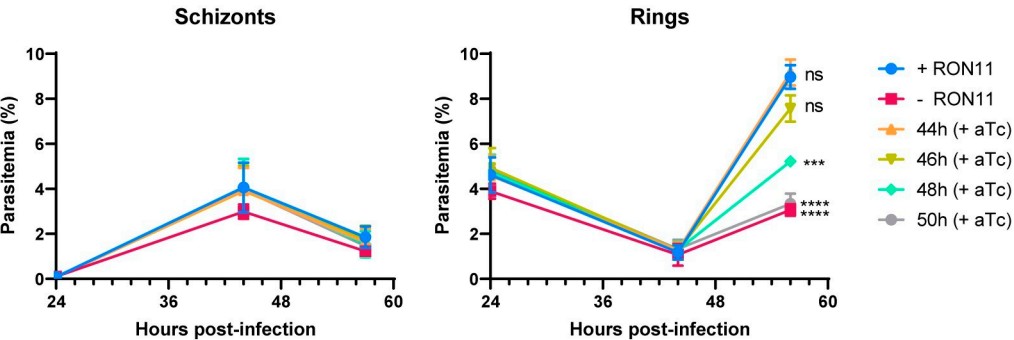

**Fig 6. RON11 knockdown phenotype is rescued by addition of aTc during the last 2 h of the asexual life cycle.**
Growth of RON11[apt] parasites in the presence or absence of aTc. Tightly synchronous parasites were grown for 44 hpi and aTc was added back every 2 h. Samples were collected at 24, 44 and 56 hpi, and measured via flow cytometry to differentiate between rings and schizonts ($n$ = 3 biological replicates; error bars = SEM; *ns = non-significant; ***$p < 0.001$; ****$p < 0.0001$ by one-way ANOVA compared to + RON11; the underlying data can be found in S1 Data). hpi, hours post-invasion.

hpi and allowed them to fully develop into segmented schizonts for another 4 h (S6 Fig). The number of rhoptries in these segmented merozoites was observed via U-ExM (S6 Fig). These data show that several merozoites within a schizont now had 2 rhoptries, while other merozoites in the same schizont still had one rhoptry (S6 Fig). Thus, the U-ExM data suggest that the restoration of growth upon addition of aTc at 46 hpi is due to the development of merozoites with 2 rhoptries (S6 Fig).

As our data shows that RON11 acts during the last hours of schizogony (Fig 6), we hypothesized it might have a role in regulating the formation of the last rhoptry pair during late schizogony. To test our hypothesis, we employed U-ExM in combination with NHS-ester to follow the development of rhoptry biogenesis during schizogony in the presence and absence of RON11. It has been previously shown that during their biogenesis, rhoptries associate with the branches of the outer CPs in a ratio of 1 rhoptry per branch during the early mitotic events of schizogony [25,27,28] (Fig 7A). Once merozoite segmentation starts, each outer CP branch accommodates 2 rhoptries, which results in the final rhoptry pair within each merozoite [27] (Fig 7A). These events can be tracked using the NHS-ester stain, which stains the CPs and rhoptries during schizogony [27,46]. By analyzing at 44 to 48 h synchronized schizonts, we found that the formation and segregation of rhoptries during the initial mitotic events was not inhibited in -RON11 parasites (Fig 7B and 7C). Surprisingly, we discovered that the loss of RON11 impaired the de novo formation of the second rhoptry after the final mitotic event (Fig 7D). Initiation of merozoite segmentation on those events was confirmed by the presence of basal complex rings in association with the CP/rhoptry-complex densities (Figs 7 and S7). Altogether, these data confirm our thesis that RON11 regulates the de novo biogenesis of the second rhoptry after the last mitotic event during schizogony (Fig 7).

## Discussion

Similar to most rhoptry proteins studied to date in the asexual RBC stages [21], we have demonstrated that RON11 is indispensable for parasite development, particularly merozoite invasion of RBCs. This aligns with previous failed attempts to generate RON11 knockouts in *P. berghei* parasites which supported essentiality of RON11 in blood stages [29,30]. We used a combination of live cell video microscopy cytochalasin D-based attachment assays to show that initial engagement with RBC occurs normally in absence of RON11 but merozoites fail to

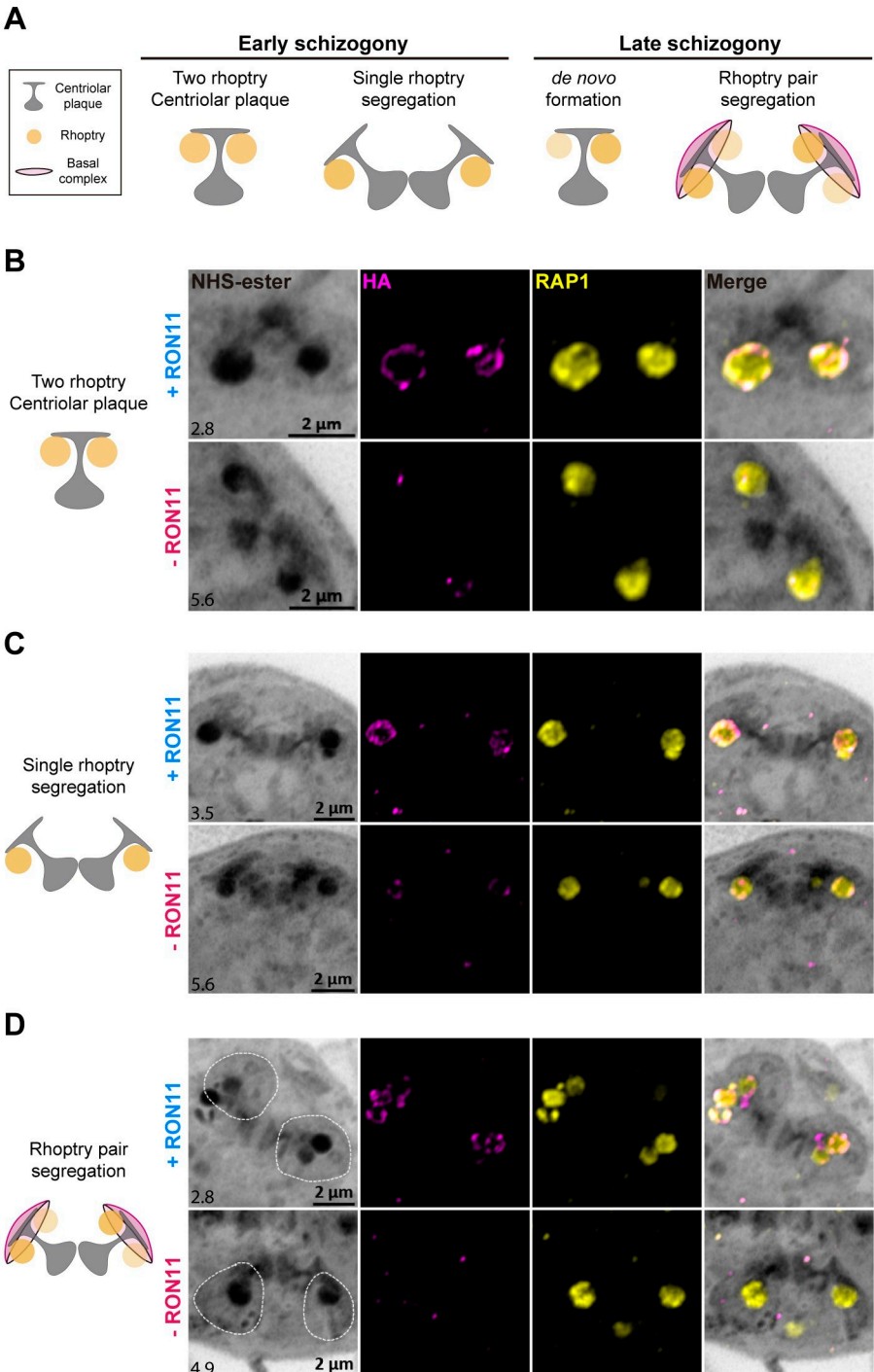

**Fig 7. RON11 is required for the de novo biogenesis of the final rhoptry pair during merozoite segmentation. (A)** Schematic of the current model for de novo formation of rhoptries during schizogony. **(B–D)** Representative U-ExM images of different steps of rhoptry biogenesis in RON11[apt] late schizonts in the presence or absence of aTc. **(B)** Two rhoptries associated to each branch of a CP in the presence or absence of aTc. **(C)** Single rhoptries being segregated with each branch of a dividing CP during mitosis, and **(D)** final rhoptry pairs being segregated with a CP branch during the final mitotic event. Basal complexes are demarcated by a discontinuous line. Late-schizont stage parasites were stained with NHS-Ester (grayscale), anti-HA (magenta), and anti-RAP1 (yellow). Selected Z stack images were projected as a combined single image. Number on image = Z-axis thickness of projection in μm. CP, centriolar plaque; HA, hemagglutinin; NHS, N-hydroxysuccinimide; U-ExM, ultrastructure expansion microscopy.

enter into RBCs. While the actomyosin motor and its components are well established as being involved in merozoite internalization [12,14], no rhoptry protein has been associated with a role in this final stage of invasion. Interestingly, we observed by live imaging that -RON11 merozoites that are unable to complete invasion, are still able to induce strong deformations on RBCs. Since gliding motility is required for deformation [5,15], this suggests that RON11 may not be required for motility of *P. falciparum* merozoites, unlike what was previously observed for *P. berghei* sporozoites [31]. RON11-deficient merozoites engage and deform host cells for significantly longer before the RBCs undergo echinocytosis, a process that is thought to follow rhoptry secretion [6]. Therefore, RON11 may not be essential for rhoptry discharge but may play a role in the efficiency or timing of rhoptry secretion. In agreement with this observation, -RON11 merozoites in the presence of cytochalasin D proceeded to attach and secrete RAP1 into RBCs.

Following the kinetics of invasion by live cell microscopy also allows us to surmise the step of invasion where RON11 functions. Previous studies using this approach revealed key receptor–ligand interactions as molecular determinants of specific steps in merozoite invasion [6]. As -RON11 merozoites are able to attach, and deform RBCs, it rules out a role for RON11 in these early steps. Following deformation, the RH5-basigin interaction leads to an opening at the interface of the merozoite and RBC initiating the secretion of rhoptry neck/bulb proteins. Blocking this interaction leads to normal deformation but prevents rhoptry secretion and no echinocytosis [6]. Finally, the AMA1-RON2 interaction mediates tight junction formation that provides an anchor and allows the actomyosin motor to drive the merozoite inside the RBC [22]. Peptide inhibitors of this interaction lead to strong deformation and echinocytosis, a result of rhoptry secretion [6]. Interestingly, RON11 knockdown resulted in merozoites unable to internalize but stay attached to the RBC and induce echinocytosis. This suggests that RON11 perhaps functions downstream of the RH5-basigin interaction. Furthermore, deformation times were significantly longer with -RON11 merozoites compared to +RON11 merozoites (average of 21 s and 6.5 s, respectively). Rhoptry secretion coincides with internalization and immediately follows the deformation step. The longer duration of deformation may therefore reflect a temporal defect in rhoptry secretion in the absence of RON11. These data are consistent with IFA data that show that RON11 is not required for the secretion of the rhoptry bulb protein, RAP1. However, since IFA uses fixed cells, we cannot determine the kinetics of rhoptry secretion. Thus, it is possible that RON11 may play a role in ensuring timely secretion of rhoptry contents after the formation of the tight junction.

Calcium signaling plays a critical role during merozoite invasion. Following RBC deformation and immediately before internalization a prominent calcium signal is observed at the apical end of the invading merozoite, suggesting a potential role for calcium sensing protein(s) at this step of invasion [6]. This signal occurs downstream of RH5-basigin interaction and is not inhibited by AMA1-RON2 blocking peptide agonists [6]. It is possible that RON11 through its C-terminal EF hand motifs, which could bind calcium, may play a role in regulating the kinetics of rhoptry secretion and/or subsequent steps in invasion.

Rhoptries are critical secretory organelles conserved among apicomplexans, with differences in numbers depending on the species as well as between different life cycle stages. During *P. falciparum* intraerythrocytic development, rhoptries are initially observed forming de novo during early schizogony [27]. Then, rhoptries replicate throughout multiple mitotic events until each merozoite contains 2 rhoptries by the end of the cycle. In contrast, *T. gondii* can have as many as 8 to 12 rhoptries, but only 2 are known to be docked at the apical end during invasion [17]. Rhoptry biogenesis remains a poorly understood process, primarily due to the small size of *P. falciparum*'s rhoptries, which, coupled with the limited resolution offered by conventional microscopy, has made their study challenging. Recent advances in microscopy

techniques, such as U-ExM, have opened new possibilities for exploring these micron-sized secretory organelles [47]. Using U-ExM, we observed that RON11 is required for the proper biogenesis of rhoptry pairs, as its absence led to merozoites with single rhoptries, an unprecedented phenotype that has not been observed in any other apicomplexan system. Only 3 parasite proteins, Sortilin, RAMA, and CERLI2, have been shown to have a role in the biogenesis of rhoptries in *P. falciparum* [39,48,49]. None of them share any structural similarities, localize to the same region within rhoptries, or generate a knockdown phenotype analogous to RON11. For instance, RAMA and CERLI2 loss causes morphological alterations in the rhoptry necks [39,49], while RON11's single rhoptries did not display any apparent structural change at the resolution of U-ExM imaging. Our data strongly suggests that RON11 is required for de novo rhoptry biogenesis at the end of schizogony, when daughter merozoites are being formed via cytokinesis. This thesis is supported by our initial observations by U-ExM that rhoptries are formed and segregated with the CPs without issues in -RON11 parasites during the early asynchronous mitotic events of schizogony. These observations align with our findings that -RON11 parasite growth is rescued by reversing the knockdown as late as 46 h post invasion, in the 48-h asexual life cycle. The U-ExM data showing individual merozoites with 1 or 2 rhoptries within a single rescued schizont are also consistent with prior observations that show that merozoite development during schizogony occurs asynchronously [28,50].

In contrast, we observed that the de novo formation of the initial rhoptry was not disrupted in -RON11 parasites during schizogony. However, during cytokinesis, we noticed that one rhoptry was being distributed along with the dividing CPs into a nascent -RON11 merozoite, instead of 2 rhoptries as observed in +RON11 parasites. These observations are in agreement with our findings of a roughly 50% reduction in the number of rhoptries per schizont. Interestingly, we observed that RON11 knockdown also led to about a 50% reduction in the quantity of other rhoptry proteins, which is consistent with the finding that only half the amount of rhoptries is formed. This is a somewhat surprising result as prior work on other *P. falciparum* organelles suggest that a defect in organelle biogenesis does not lead to a loss in expression of organellar proteins. For example, upon loss of the parasite plastid known as the apicoplast, proteins targeted to the apicoplast are synthesized and appear to be stuck in secretory vesicles [51–53]. Our data suggest that RON11 may function in a pathway that regulates synthesis of rhoptry proteins, either via transcription or translation. These data suggest that a signaling pathway may exist that ensures synthesis of rhoptry proteins only when rhoptry biogenesis occurs. Future transcriptomic and proteomic studies should resolve the role of RON11 in this de novo biogenesis pathway. Since RON11 has a pair of calcium-binding EF-hand domains predicted to be in the cytoplasm, it raises the possibility that a calcium-dependent pathway may function in biogenesis of rhoptries. During the final hours of schizogony, calcium-dependent pathways also trigger egress of the merozoites from the host RBC [16]. This study raises the possibility that a similar calcium signal may also kickstart, via RON11, the de novo biogenesis of the second rhoptry to ensure that the egressing merozoites have 2 rhoptries each for invading the next host RBC. It is unlikely that these are the same calcium signals because blocking the egress-specific calcium signal using PKG inhibitors does not prevent rhoptry biogenesis.

Since -RON11 merozoites with a single, functional rhoptry attach and secrete their contents into the host RBC but fail to internalize into the host cell, this raises the question: do *P. falciparum* merozoites need 2 rhoptries to invade host RBCs? One possibility is that since the single rhoptry -RON11 merozoites have half as much rhoptry contents compared to the +RON11 merozoites, the failure to internalize may be due to insufficient amounts of invasion ligands such as RON2. It is possible that while RAP1 is secreted from the single rhoptry in -RON11 parasites, secretion of other rhoptry neck proteins is disrupted or the timing of rhoptry

secretion is disrupted in the absence of RON11. Distinguishing between these models will require developing tools to observe rhoptry secretion in live parasites. Another possibility is that RON11 has another biogenesis-independent function in merozoite invasion. Other rhoptry proteins have been shown to have multiple independent functions [39,54–56]. Our data suggests that RON11 may function in merozoite invasion because rhoptry proteins do localize correctly in -RON11 parasites, they are processed accurately by PMIX, and anti-RON11 antibodies inhibit intraerythrocytic growth of *P. falciparum*.

Together, these data suggest a model where RON11 has 2 functions in the final hours of the intraerythrocytic life cycle of *P. falciparum*. During segmentation of the schizont, RON11 is required to trigger the de novo biogenesis of the final rhoptry pair, which ensures each merozoite ends up with 2 rhoptries. After the merozoite makes contact with a new host RBC, RON11 is required for internalization into the new host cell. Calcium signals play a critical role in both merozoite egress as well as invasion. It remains to be seen if the calcium-binding EF-hand domains of RON11 play a role in either its function in rhoptry biogenesis or its function in merozoite invasion. The discovery of the biological function of RON11 now provides an avenue to investigate the molecular mechanisms driving rhoptry biogenesis and how it is linked to merozoite invasion.

## Materials and methods

### Construction of RON11 plasmids

Genomic DNA was isolated from *P. falciparum* NF54$^{attB}$ cultures using the QIAamp DNA blood kit (Qiagen). PCR products were inserted into the respective plasmids using the NEBuilder HiFi DNA Assembly system (NEB). All constructs used in this study were confirmed by sequencing. All primers used in this study are in S1 Table.

For generation of the plasmid pKD-RON11-Apt, sequences of approximately 500 bp of homology to the RON11 (Pf3D7_1463900) C-terminus and 3′ UTR were amplified using primer pairs P3-P4 and P5-P6, respectively (S1 Table). Amplicons were then inserted into pKD [33] digested by with AatII and AscI. For expression of a RON11 gRNA, oligo P7 was inserted into cut pUF1-Cas9 (S1 Table).

### Parasite culture and transfections

*Plasmodium* parasites were cultured in RPMI 1640 medium (NF54$^{attB}$, RON11$^{apt}$) supplemented with AlbuMAX I (Gibco), and transfected as described earlier [57].

For generation of RON11$^{apt}$ parasites, the pKD-RON11-Apt plasmid (20 μg) and the respective pUF1-Cas9 plasmid (50 μg) were transfected into NF54$^{attB}$ parasites in duplicate. Before transfection pKD plasmids were digested overnight with EcoRV (NEB). The enzyme was then subjected to heat inactivation for 20 min at 65˚C and then mixed with the pUF1-Cas9-RON11gRNA plasmid. Transfected parasites were grown in 0.5 μm anhydrous tetracycline (aTc) (Cayman Chemical). Drug pressure was applied 48 h after transfection, using blasticidin (BSD) (Gibco) at a concentration of 2.5 μg/ml, selecting for pKD-RON11-Apt expression. After parasites grew back from transfection, integration was confirmed by PCR, and then cloned using limiting dilution. Clones were maintained in mediums containing 0.5 μm aTc and 2.5 μg/ml BSD.

### Growth assays

For all assays, aliquots of parasite cultures were incubated in 8 μm Hoechst 33342 (Thermo Fisher Scientific) for 20 min at room temperature and then fluorescence was measured using a

CytoFlex S (Beckman Coulter) flow-cytometer. Flow cytometry data were analyzed using FlowJo software (Tree Star) and plotted using Prism (GraphPad Software).

For the growth assay, synchronous schizont-stage parasites were washed 5 times with RPMI 1640 medium and split into 2 cultures, one resuspended in medium containing 0.5 μm aTc and 2.5 μg/ml BSD (+RON11), and the other one in medium containing only 2.5 μg/ml BSD (-RON11). Cultures were then transferred to a 96-well plate at 0.2% parasitemia and grown for 6 days. Parasitemia was monitored every 48 h.

For the invasion versus egress assay, synchronous ring-stage parasites were washed and split into 2 cultures with and without aTc, as previously described. Parasites were then allowed to develop into mature schizonts to be isolated using a Percoll gradient (Genesee Scientific). Enriched parasites were then incubated for 4 h at 37˚C in pre-warmed RPMI media supplemented with the PKG inhibitor, ML10 compound (25 nM) (obtained from S. Osborne, BEI resources) [40]. After incubation, parasites were washed twice with pre-warmed RPMI media and transferred immediately to pre-warmed RBCs at 2% hematocrit. Parasitemia was monitored every 2 h, for 8 h.

For the rescue assay, parasites were synchronized to a 1-h window. Synchronized ring-stage parasites were washed and split into 6 cultures in a 6-well plate (+ RON11,—RON11, 44 h, 46 h, 48 h, and 50 h). Culture "+RON11" was grown in the presence of aTc, while the others were grown in absence of aTc. Parasites were allowed to develop into schizonts and aTc was supplemented back into cultures at their corresponding time point, except for the "-RON11" culture. Parasitemia was monitored at 44 and 56 hpi.

## Western blotting

Parasites were synchronized into ring stages, washed 6 times in RPMI media, and then split into 2 cultures with and without aTc, as previously described. Synchronous parasites were then allowed to develop into mature schizonts to be then incubated with E64 for 4 h at 37˚C. After treatment, culture pellets were treated with ice-cold 0.04% saponin in 1× PBS to isolate parasites from host cells. The parasite pellets were subsequently solubilized in protein loading dye with Beta-mercaptoethanol (LI-COR Biosciences) and used for SDS-PAGE.

Primary antibodies used in this study included mouse-anti-HA (6E2; Cell Signaling Technology; 1:2,000), rabbit-anti-PfEF1α (from Daniel Goldberg, 1:2,000), mouse-anti-RAP1 (2.29; from Jana McBride via the European Malaria Reagent Repository; 1:500) [58], mouse-anti-RON4 (10H11; from Alan Cowman; 1:500) [59], mouse-anti-MSP1 (12.4; from Jana McBride via the European Malaria Reagent Repository; 1:500) [60]. Secondary antibodies used were IRDye 680 CW goat-anti-rabbit IgG and IRDye 800CW goat-anti-mouse IgG (Li-COR Biosciences; 1:20,000). Membranes were imaged using the Odyssey Clx Li-COR infrared imaging system (Li-COR Biosciences). Images were processed and analyzed using ImageStudio (Li-COR Biosciences).

## Microscopy and image analysis

For all microscopy assays, parasites were synchronized, washed, and split into 2 cultures with and without aTc, as previously described.

For IFAs, cells were fixed following the previously described protocol [61].

For localization assays, schizont-stage parasites were smeared onto a slide, fixed, and permeabilized with acetone for 10 min at room temperature. Fixed slides were then washed 3 times in 1× PBS for 5 min each.

For rhoptry secretion assays, synchronous parasites were allowed to develop into mature schizonts to be isolated using a Percoll gradient (Genesee Scientific). Enriched parasites were

then incubated for 4 h at 37˚C in pre-warmed RPMI media supplemented with 25 nM ML10 compound. After incubation, parasites were washed twice with pre-warmed RPMI media and transferred immediately to RBCs at 1% hematocrit. Cultures were then allowed to egress for 30 min at 37˚C in media supplemented with 1 μm cytochalasin D (Invitrogen). After incubation, parasites were fixed with 4% paraformaldehyde (PFA) (Electron Microscopy Sciences) and 0.03% glutaraldehyde, before being permeabilized with 0.1% Triton-X100.

For all IFA's, fixed and permeabilized cells were blocked for 1 h using 3% BSA in PBS. After blocking, samples were incubated with primary antibodies in 3% BSA overnight at 4˚C. Following primary antibody incubation, samples were washed 3 times in 1× PBS for 10 min before incubation with secondary antibodies in 3% BSA in PBS for 2 h. After secondary antibody staining, samples were washed 3 times in 1× PBS, to be then mounted.

Primary antibodies used in the IFAs included rabbit-anti-HA (SG77; Thermo Fisher Scientific; 1:100), mouse-anti-RAP1 (2.29; from Jana McBride via the European Malaria Reagent Repository; 1:500) [58], mouse-anti-RON4 (10H11; from Alan Cowman; 1:200) [59]. Secondary antibodies used were Alexa Fluor 488 and Alexa Fluor 546 (Life Technologies, 1:1,000).

After mounting the cells using ProLong Diamond with 4′,6′-diamidino-2-phenylindole (DAPI) (Invitrogen), they were imaged using a DeltaVision II microscope system with an Olympus Ix-71 inverted microscope. Images were collected as a Z-stack and deconvolved using SoftWorx (GE HealthCare), then displayed as a maximum intensity projection. Adjustments to brightness and contrast were made for display purposes using Adobe Photoshop. Colocalization was analyzed using the Pearson's correlation coefficient (PCC) calculated by the Cell Profiler software (Broad Institute).

For growth-tracking assays, synchronous schizont-stage parasites were washed 5 times and split into 2 cultures, one without and one with aTc. Parasites were allowed to develop during 2 life cycles and sample aliquots were collected at different time points (0, 4, 26, 46, 50, and 54 h post wash), starting with schizonts.

For attachment assays, parasites were collected as previously described in the rhoptry secretion assay with a slight change. After ML10 removal, schizont-stage parasites were allowed to egress during 2 h at 37˚C in the presence and absence of 1 μm cytochalasin D. Sample aliquots were collected after incubation.

For both assays, aliquots were smeared into glass slides, fixed and stained using Hema3 solutions (PROTOCOL, Fisher Healthcare), and then analyzed by light microscopy. Slides were imaged using a Zeiss Axio Scope A1 microscope with a Zeiss Axiocam 305 color camera. Images were processed using Adobe Photoshop.

## Ultrastructural expansion microscopy (U-ExM)

Cultures for U-ExM were synchronized to mature schizont parasites, following the previously described methods. Ultrastructure expansion microscopy (U-ExM) was performed as described previously [46], with minor modifications.

To start, 12-mm round coverslips were treated with poly-D-lysine for 1 h at 37˚C. They were then washed 3 times with MilliQ water and placed in a 24-well plate. Parasite cultures with approximately 5% parasitemia were adjusted to 0.5% hematocrit. Then, 1 ml of parasite culture was added to the well containing the treated coverslip and incubated for 1 h at 37˚C.

After the incubation, the supernatant was carefully removed, and a fixative solution (4% v/v PFA in PBS) was added, followed by a 20-min incubation at 37˚C. The coverslips were washed 3 times with 1× PBS and incubated overnight at 37˚C in 500 μl of 1.4% formaldehyde/2% acrylamide (FA/AA) in PBS.

The monomer solution (19% sodium acrylate, 10% acrylamide, 0.1% N,N′-methylenebisa-crylamide in PBS) was prepared a day prior and stored at −20°C. Before gelation, coverslips were removed from FA/AA solution and washed 3 times in 1× PBS.

For gelation, 5 μl of 10% tetramethylenediamine (TEMED) and 5 μl of 10% ammonium persulfate (APS) were added to 90 μl of the monomer solution, briefly vortexed, and 35 μl of the monomer mixture were pipetted onto parafilm. The coverslips were placed on top with the cell-side facing down, and the gels were incubated at 37°C for 30 min.

Next, the gels were transferred into a 6-well plate containing denaturing buffer (200 mM sodium dodecyl sulfate (SDS), 200 mM NaCl, 50 mM Tris, pH 9) and incubated for 15 min incubation at room temperature. Afterward, the gels were separated from the coverslips and transferred to 1.5 ml tubes with the denaturing buffer for 90-min incubation at 95°C.

Subsequently, the gels were incubated with secondary antibodies diluted in 1× PBS for 2.5 h. After denaturation, gels were transferred to Petri dishes containing 25 ml of MilliQ water and incubated 3 times for 30 min at room temperature with shaking, changing the water in between. The gels were measured and subsequently shrunk using 2 washes with 1× PBS. They were then transferred to a 24-well plate for blocking in 3% BSA in PBS at room temperature for 30 min. After blocking gels were incubated with primary antibodies diluted in 3% BSA overnight at room temperature.

Following primary antibody incubation, the gels were washed 3 times in 0.5% PBS-Tween 20 for 10 min before incubation with secondary antibodies diluted in 1× PBS for 2.5 h.

After secondary antibody staining, the gels were washed 3 times with 0.5% PBS-Tween 20. Then, gels were transferred back to 10 cm Petri dishes for the second round of expansion, involving 3 incubations with MilliQ water. After re-expansion, the gels were either imaged immediately or stored in 0.2% propyl gallate in water until imaging.

The primary antibodies used were rat-anti-HA 3F10 (Roche, 1:25), mouse-anti-RAP1 (2.29; from Jana McBride via the European Malaria Reagent Repository; 1:500) [58], mouse-anti-RON4 (10H11; from Alan Cowman; 1:200) [59]. The secondary antibodies used were Alexa Fluor 488 and Alexa Fluor 546 (Life Technologies, 1:500), NHS-ester 405 (Thermo Fisher, 1:250). The gels were imaged using a Zeiss LSM 980 microscope with Airyscan 2. Images were collected as a Z-stack, processed by Airyscan, and then displayed as a maximum intensity projection. Adjustments to brightness and contrast were made using ZEN Blue software for display purposes.

## Expression and purification of recombinant RON11

An *E. coli* codon optimized DNA fragment corresponding to amino acids 32–442 of RON11 was synthesized (Genscript) and cloned into pMAL vector (NEB) in frame with an N-terminal maltose binding protein (MBP) tag and c-terminal His6 tag. Protein expression was induced using 0.4mM IPTG at 16°C for 12 to 14 h. The cell pellet was resuspended in a lysis buffer (300 mM NaCl, 10 mM imidazole, 500 mM NaH2PO4, and 1 μm PMSF, 10 mM BME, and 0.1% Tween20). Following sonication, the soluble fraction containing recombinant MBP-RON11--His6 protein was affinity purified twice on immobilized-metal affinity chromatography (IMAC) column using FPLC (Biorad) and further purified on CHT Ceramic Hydroxyapatite column. Fractions containing rMBP-RON11 were pooled and dialyzed into phosphate-buffered saline (PBS) containing 1 μm PMSF, 10 mM BME, and 0.1% Tween20 and kept at −80°C.

## Animal immunization, IgG purification, and neutralization assay

Animal experiments were approved by the Johns Hopkins Animal Care and Use Committee (ACUC, protocol# RA22H291). Four female Sprague Dawley rats (Charles River Laboratory),

5 to 6 weeks old, were used for immunizations. Recombinant MBP-RON11 (25μg/rat/immunization) was mixed 1:1 with Addavax adjuvant and injected subcutaneously 3 times in 2-week intervals. Two weeks following the last immunization blood was collected by terminal heart bleed. Total rat IgG was purified from serum using pre-equilibrated protein G column (GE health sciences), dialyzed against RPMI 1640 medium, and sterile filtered.

Synchronized trophozoite stage 3D7 strain parasites were adjusted to 0.5% parasitemia in 4% hematocrit in a 96-well plate and incubated with anti-RON11 IgG or control rat IgG (8 mg/ml) at the indicated concentrations with or without pre-incubation with 3 μm rMBP protein. After 72 h incubation at 37˚C, cells were stained with SYBR Green I and parasitemia was measured using AttuneNxT flow cytometer (Thermo Scientific).

## Supporting information

**S1 Movie. Time-lapse imaging of RON11^apt merozoite (white arrowhead) grown with aTc for 48 h invading a host red blood cell.** Representative merozoite from RON11^apt parasites growth with aTc is shown (*n* = 3 biological replicates, 35 merozoites).
(AVI)

**S2 Movie. Time-lapse imaging of RON11^apt merozoite (white arrowhead) grown without aTc for 48 h interacting with a host red blood cell.** Representative merozoite from RON11^apt parasites growth with aTc is shown (*n* = 3 biological replicates, 53 merozoites).
(AVI)

**S1 Table. List of primers used in the study to generate the cell lines RON11^apt.**
(DOCX)

**S1 Data. Duration of echinocytosis.**
(XLSX)

**S1 Fig. RON11 knockdown quantification and localization to the parasite periphery during early-ring stages. (A)** Overexposed western blot of the knockdown of RON11 shown in Fig 1C. Lysates were collected from E64-arrested RON11apt parasites in the presence or absence of aTc. Samples were probed with antibodies against the HA tag. The protein marker sizes are shown on the left. Blot shows 4 biological replicates. **(B)** IFAs showing the localization of RON11 in RON11^apt rings with respect to the rhoptry markers RON4 and RAP1. Synchronous parasites were fixed with PFA and stained with specific antibodies. Images from left to right are DIC, DAPI (nucleus, cyan), anti-HA (RON11, magenta), anti-RON4 or RAP1 (yellow), and fluorescence merge. Z stack images were deconvolved and projected as a combined single image. Representative images of 2 biological replicates.
(TIF)

**S2 Fig. Expression and purification of recombinant RON11 and validation of anti-RON11 antibody. (A)** Expression of rMBP-RON11 (aa 32–442) protein in *E.coli*. Soluble (S) and insoluble (I) fractions from cell lysates without and without IPTG induction. M; Marker. **(B)** Western blot analysis of the soluble fraction using anti-His antibody and detected using anti-mouse HRP conjugated secondary antibody. **(C)** rMBP-RON11 following IMAC purification and subsequent clean-up using CHT column. **(D)** Western blot analysis of *P. falciparum* schizont lysates using anti-RON11 antibody. (E) IFAs showing the localization of RON11 in RON11apt mature schizonts. Images from left to right are DAPI (nucleus, cyan), anti-RON11 (magenta), anti-HA (yellow), anti-RAP1 (green), and fluorescence merge. (F) IFAs showing the knockdown of RON11 in RON11apt mature schizonts. Images from left to right are DIC, DAPI (nucleus, cyan), anti-RON11 (magenta), and fluorescence merge. Z stack images were

deconvolved and projected as a combined single image.
(TIF)

**S3 Fig. RON11$^{apt}$ merozoites attached to RBCs in the presence or absence of aTc. (A)**
Duration between first contact of merozoites with RBCs and observation of RBC deformation
($n$ = 2 biological replicates, 20 merozoites for +RON11 and 13 for -RON11; error bars = SEM;
\*$p < 0.05$ by unpaired two-tailed $t$ test; the underlying data can be found in S1 Data). **(B)**
Duration between merozoite first merozoite-induced RBC deformation and echinocytosis
($n$ = 2 biological replicates, 20 merozoites for +RON11 and 13 for -RON11; error bars = SEM;
ns = non-significant by unpaired two-tailed $t$ test; the underlying data can be found in S1
Data). **(C)** Quantification of RBCs showing echinocytosis after merozoite-induced deforma-
tion ($n$ = 3 biological replicates, 34 merozoites for +RON11 and 44 for -RON11; the underlying
data can be found in S1 Data). **(D)** Deformation scores based on the strength of merozoite-
RBC interaction. 0 = no deformation; 1 = shallow indentation/membrane pinching; 2 = deeper
indentation to the side of RBC/intermediate level of host cell membrane pinching around the
parasite. ($n$ = 3 biological replicates, 44 merozoites for each condition; ns = non-significant by
chi-squared test; the underlying data can be found in S1 Data). **(E)** Schematic of the actin-poly-
merization inhibitor assay. Schizonts were tightly synchronized incubating for 4 h with the
PKG inhibitor, Compound 1. After incubation, schizonts were washed twice and then trans-
ferred to fresh red blood cells in the presence of the actin inhibitor, cytochalasin D, for 30 min.
**(F and G)** Quantification of RON11$^{apt}$ merozoites attached to RBCs in the presence or absence
of aTc, after incubation with **(F)** or without **(G)** cytochalasin D. Parasites were smeared 30
min after Compound 1 removal, stained with Hema 3, and scored by light microscopy.
Attached-merozoites were blindly scored and represented as the percentage of events per 100
RBCs ($n$ = 5 and 3 biological replicates, respectively; error bars = SEM; ns = non-significant;
\*\*$p < 0.01$ by unpaired two-tailed $t$ test; the underlying data can be found in S1 Data).
(TIF)

**S4 Fig. RON11 knockdown can generate multiple rhoptries but does not affect the number
of merozoites. (A)** Representative images of RON11$^{apt}$ parasite showing multiple rhoptries
accumulating within fully developed schizonts in the absence of aTc. ML10-arrested parasites
were expanded by U-ExM, fixed with PFA, and stained with NHS-Ester (grayscale), anti-HA
(magenta), and the DNA dye SYTOX (cyan). Selected Z stack images were projected as a com-
bined single image. Number on image = Z-axis thickness of projection in μm. **(B and C)**
Quantification of **(B)** rhoptries per merozoites and **(C)** merozoites per schizont in the presence
and absence of aTc. Merozoites were scored based on the number of nuclei observed ($n$ = 4
biological replicates, 28 schizonts for +RON11 and 36 for -RON11; error bars = SEM;
ns = non-significant; \*\*\*$p < 0.001$ by unpaired two-tailed $t$ test; the underlying data can be
found in S1 Data).
(TIF)

**S5 Fig. RON11 knockdown does not impact the expression and secretion of MSP1 and
AMA1. (A)** Representative IFAs of RON11$^{apt}$ schizonts showing secretion of MSP1 proteins
in the presence or absence of aTc. Images from left to right are phase-contrast, DAPI (nucleus,
blue), anti-HA (RON11, green), anti-MSP1 (red), and fluorescence merges. Z stack images
were deconvolved and projected as a combined single image. Representative images of 3 bio-
logical replicates. **(B)** (Top) Western blot of parasite lysates isolated from E64-arrested
RON11$^{apt}$ parasites in the presence or absence of aTc. Samples were probed with antibodies
against MSP1 and EF1α (loading control). The protein marker sizes are shown on the left.
Representative blot of 4 biological replicates shown. (Bottom) Quantification of MSP1 in

E64-arrested RON11^apt parasites in the presence or absence of aTc. Band intensities were normalized to the loading control, EF1α (*n* = 4 biological replicates; error bars = SD; ns = non-significant by unpaired two-tailed *t* test; the underlying data can be found in S1 Data). **(C)** Representative IFAs of RON11^apt schizonts showing AMA1 localization in the presence or absence of aTc. Images from left to right are DIC, DAPI (nucleus, blue), anti-HA (RON11, green), anti-AMA1 (red), and fluorescence merges. Z stack images were deconvolved and projected as a combined single image. Representative images of 3 biological replicates.
(TIF)

**S6 Fig. Addition of aTc at 46-h schizonts partially recovers the single-rhoptry phenotype.** Representative U-ExM images of E64-treated late RON11^apt schizonts after supplementing back aTc at 46 hpi. Late-schizont stage parasites were stained with NHS-Ester (grayscale) and anti-RAP1 (yellow). Selected Z stack images were projected as a combined single image. Number on image = Z-axis thickness of projection in μm.
(TIF)

**S7 Fig. RON11 is required for the de novo biogenesis of the final rhoptry pair during merozoite segmentation.** Representative images of different steps of rhoptry biogenesis in RON11^apt late schizonts in the presence or absence of aTc. Basal complexes are demarcated by a discontinuous white line. The top panel shows the start of merozoite segmentation by the basal complex and each successive panel shows later stages of segmentation. Late-schizont parasites were expanded by U-ExM, fixed with PFA, and stained with NHS-Ester (grayscale), anti-HA (magenta), and anti-RAP1 (yellow). Selected Z stack images were projected as a combined single image. Number on image = Z-axis thickness of projection in μm.
(TIF)

**S1 Raw Images. Original blots.**
(PDF)

## Acknowledgments

We thank Dan Goldberg for anti-EF1α, Alan Cowman for anti-RON4, The European Malaria Reagent Repository for anti-RAP1 antibodies; Julie Nelson and Juan Bustamante at the CTEGD Cytometry Shared Resource Laboratory for help with flow cytometry and analysis; and Muthugapatti Kandasamy at the Biomedical Microscopy Core at the University of Georgia for help with microscopy.

## Author Contributions

**Conceptualization:** David Anaguano, Vasant Muralidharan.

**Data curation:** David Anaguano, Manuel A. Fierro, Vasant Muralidharan.

**Formal analysis:** Prakash Srinivasan, Vasant Muralidharan.

**Funding acquisition:** Prakash Srinivasan, Vasant Muralidharan.

**Investigation:** David Anaguano, Opeoluwa Adewale-Fasoro, Grace W. Vick, Sean Yanik, Manuel A. Fierro, Prakash Srinivasan, Vasant Muralidharan.

**Methodology:** David Anaguano, Opeoluwa Adewale-Fasoro, Grace W. Vick, James Blauwkamp, Sabrina Absalon.

**Project administration:** Vasant Muralidharan.

**Resources:** David Anaguano, Sean Yanik, James Blauwkamp, Sabrina Absalon, Prakash Srinivasan, Vasant Muralidharan.

**Supervision:** Sabrina Absalon, Prakash Srinivasan, Vasant Muralidharan.

**Validation:** David Anaguano, Opeoluwa Adewale-Fasoro, Grace W. Vick.

**Visualization:** David Anaguano, Vasant Muralidharan.

**Writing – original draft:** David Anaguano, Vasant Muralidharan.

**Writing – review & editing:** David Anaguano, Sabrina Absalon, Prakash Srinivasan, Vasant Muralidharan.

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
