## [Editor Report · Decision Letter 0]

28 Feb 2024

Dear Dr Muralidharan, 

Thank you for submitting your manuscript entitled "Plasmodium RON11 triggers biogenesis of the merozoite rhoptry pair and is essential for erythrocyte invasion" for consideration as a Research Article by PLOS Biology. I would like to apologize for the delay in sending you the initial decision. 

Your manuscript has now been evaluated by the PLOS Biology editorial staff, as well as by an academic editor with relevant expertise, and I am writing to let you know that we would like to send your submission out for external peer review.

Once your full submission is complete, your paper will undergo a series of checks in preparation for peer review. After your manuscript has passed the checks it will be sent out for review. To provide the metadata for your submission, please Login to Editorial Manager (https://www.editorialmanager.com/pbiology) within two working days, i.e. by Mar 01 2024 11:59PM.

Kind regards,

Melissa

Melissa Vazquez Hernandez, Ph.D.

Associate Editor

PLOS Biology

---

## [Decision Letter · Decision Letter 1]

5 Apr 2024

Dear Vasant,

Thank you for your patience while your manuscript "Plasmodium RON11 triggers biogenesis of the merozoite rhoptry pair and is essential for erythrocyte invasion" went through peer-review at PLOS Biology. Your manuscript has now been evaluated by the PLOS Biology editors, an Academic Editor with relevant expertise, and by three independent reviewers, one of them being Robert W. Moon.

As you will see in the reports, all reviewers are positive about the relevance of the work, but still some concerns should be addressed prior to publication. Reviewer #2 and 3 request additional control in the Western Blots. Additionally, Reviewer #2 suggests additional experiments such as parasite invasion quantification and digitonin/proteinase K protection assays. All reviewers ask for adjustments in the text and clarification on some experiments.

IMPORTANT: While the experimental requests from reviewer #2 will strengthen the conclusion of the study, we leave to your discretion to perform them. Additionally, the Academic Editor has provided additional comments regarding some overstatements of the findings (see the foot of this email). Addressing the concerns of all reviewers, as well as of the Academic Editor, is essential for further consideration of your manuscript for publication in PLOS Biology.

**IMPORTANT - SUBMITTING YOUR REVISION**

*Resubmission Checklist*

*Published Peer Review*

*PLOS Data Policy*

*Blot and Gel Data Policy*

Sincerely,

Melissa

Melissa Vazquez Hernandez, Ph.D.

Associate Editor

PLOS Biology

REVIEWERS' COMMENTS

Reviewer #1: 

This study by Anaguano et al. investigates the role of Rhoptry Neck Protein 11 (RON11) in Plasmodium falciparum, the causative agent of malaria, with a focus on its involvement in erythrocyte invasion and rhoptry biogenesis. Utilizing conditional mutants of RON11, the study reveals its indispensable nature for parasite growth, primarily through its facilitation of merozoite invasion into red blood cells (RBCs). Remarkably, the loss of RON11 results in a unique phenotype where merozoites possess only one rhoptry each, impeding successful invasion despite initial attachment to RBCs and eventual secretion of rhoptry contents. Their observation suggests a novel function for RON11 in triggering the de novo formation of the second rhoptry during merozoite segmentation, shedding light on the intricate regulation of rhoptry biogenesis. The study hints at a potential role for RON11 in regulating the synthesis and secretion of rhoptry proteins, possibly mediated through calcium signaling pathways. In my view, this study provides crucial evidence and avenues to understand, and eventually elucidate, a longstanding question of why P. falciparum merozoites need 2 separate rhoptries and only fuse them upon secretion, and whether there are differences between the two rhoptries. Further studies stimulated by this work will hopefully address these important questions.

The manuscript is well-written, and conclusions are well-supported by multifaceted experiments. The findings advance our understanding of Plasmodium invasion mechanisms and pave the way for further exploration into the biology of the important organelle, the rhoptry, in apicomplexan parasites. I would like to congratulate the authors on this outstanding piece of work.

Specific comments: 

1) Line 273: "...which is thought to be indicative of secretion of contents from rhoptries into the RBC" - needs a reference. 

2) I have trouble understanding the plots in Fig. 4B. The description is "Quantification of merozoites with single, dual, or multiple rhoptries in the presence and absence of aTc." Are each dot on the plot representing one merozoite? If so, what does the Y-axis "percentage of merozoites" mean for each dot? 

3) Line 543: "…only two are known to release their contents during invasion (Mageswaran et al. 2021)" - the referenced paper showed evidence for two rhoptries being ready for secretion at a time, but it's not concluded if only two rhoptries are secreted for invasion. Please revise.

Reviewer #2: 

Red blood cell invasion of the malaria parasite is an essential, highly unique and specialized process. This tightly coordinated cell-cell interaction relies on secretory organelles including the micronemes, rhoptries and dense granules that secrete their contents to facilitate interactions. Rhoptries are the largest and most prominent of these invasive organelles. They are very likely Golgi-derived and manifest as dual-club-shaped organelles located at the apical pole of the invasive stage of the parasite. They are formed de novo during cytokinesis, are storing a subset of secreted proteins and secrete them during host cell invasion. Despite the essential role of these organelles for parasite proliferation, rhoptry biogenesis and function is not well understood. 

The authors of this nice manuscript are investigating the function of a multi-transmembrane rhoptry protein termed Rhoptry Neck Protein 11 (RON11). By generating conditional RON11 mutants using the tetR-DOZI aptamer system, they show that the knockdown of RON11 interferes with efficient red blood cell invasion and therefore blocks parasite proliferation. They show that RON11 deficient merozoites are able to engage with RBC and release their rhoptry content but are prevented from a productive invasion process. The authors show that the loss of RON11 leads to a decrease in the amount of rhoptry proteins. Amazingly, this goes with a drastic morphological change of the rhoptries which are reduced to a single one. Nevertheless, this "freak" rhoptry appears to be perfectly orientated within the complex architecture of the apical tip of the parasites and apparently accommodates a set of rhoptry proteins that are normally processed. They conclude that RON11 triggers the de novo biogenesis of the second rhoptry and that is apparently necessary for RBC invasion. 

This is a well-written manuscript with a thorough functional analysis of a gene coding for the multi-transmembrane rhoptry RON11 that was shown in Toxoplasma not to be essential for host cell invasion. Using a conditional knock-down system in P. falciparum the authors convincingly show that RON11 plays an important in parasite RBC invasion and present a remarkable single rhoptry phenotype.

A couple of remarks: 

1. According to the growth curve in 1D, they observed a clear growth defect only after the second cycle without aTC. It is not clear when they collected the samples to corroborate the knock-down by WB. Was it after one or two cycles without aTC? How old were the schizonts that they analyzed for invasion and microscopy experiments, do they have to keep them for two cycles without aTC to see the effect of knock-down? The information might be included somewhere but this reviewer could not find it and it appears to be important to state this upfront to allow the reader the assessment of these data points. 

2. Fig. 2: Please quantify parasite invasion (number of rings per ruptured schizonts- compared with control). Any egress (percentage of ruptured schizonts compared to control) phenotype?

3. The authors claim that that the N terminus of RON11 is accessible to antibodies during merozoite invasion and may interfere with its function in RBC invasion. What is the basis for this? Would be nice if this implied topology would be confirmed with some experimental data such as digitonin/ proteinase K protection assays?

4. The authors observed that RON11 knockdown also led to about a 50% reduction in the amount of rhoptry proteins in general - likely because of merozoites with a single rhoptry. As they claimed, this is unexpected, because a defect in organelle biogenesis should not affect protein expression. This reduction is not really fully supported from the WBs shown, although they quantified by densitometry in several experiments. They used EF1 as loading control but a loading control with a similar time of rhoptry protein expression (RON2, RAP1, AMA1, …) would be more appropriate to exclude a potential developmental delay.

5. Fig 6. - rescued RON11 knockdown phenotype by addition of aTC: Please provide data on the morphology of their rhoptries and on their protein content. 

6. Discussion: It might be advisable that the authors summarize the data they provided and align them with the two different putative functions they hypothesise: i) "RON11 triggers the de novo biogenesis of the second rhoptry" ii) "RON11 through its C-terminal EF hand motifs, which can bind calcium, may play a role in regulating the kinetics of rhoptry secretion and/or subsequent steps in invasion. While the first suggestion is supported by the mesmerizing phenotype, for the second function the data are sparse.

7. Just out of interest - why an HA and not GFP tag? 

Reviewer #3: 

This report details a fascinating analysis of the role of the malarial rhoptry neck protein, RON11. The protein is found across the apicomplexa and previous indirect evidence has suggested that it may play a critical role during invasion. Here the authors use a conditional knockdown approach to demonstrate that the protein is required for both rhoptry biogenesis and invasion - with a unique and very specific KD phenotype of production of merozoites with only a single rhoptry as opposed to the standard pair. The work is elegant, methodical, and very well written. The work also provides a unique and novel insight into two key areas of invasion and organellar biogenesis, which will be of quite wide interest to those interested in host pathogen interactions. My comments are relatively minor corrections to wording with a few missing controls which could hopefully be added without too much additional work.

L62 - This phrasing might be read to imply that invasion as a target is in use within a clinical setting, which it is not. There is of course strong interest in developing invasion targets. Rephrase sentence accordingly, also not sure González-Sanz, Berzosa, and Norman even mention invasion targets.

Line 147, a KD of 97% of protein is quoted, but there doesnt seem to be a band at all in -aTc HA land of 1C. How was the 97% calculated? If a longer exposure was used, then this one should be used for fig 1C.

Figure 1A schematic primers are labelled incorrectly. In the integration line I beleive it should read P1 P2 then P6 in order to match with Fig 1B (or Fig B could be changed to match 1A and the P6 in parental line changed to P2). 

Could the authors also confirm that P1 is situated outside of sequences used in the template plasmid? Perhaps it could be moved slightly on schematic to make this more clear (important so that P1 +P6 cant create product with episome).

Figure 1 Legend - This could be edited down a bit to be more concise. Some of the others could do with a bit of a trim too.

Figure 2C should have a control IgG, either rat preimmune or a rat IgG control.

Supplemental figure 2- IFA data comparing HA and RON11 antibody specificity is convincing, but also demonstrating loss of signal in ATC- cultures would have been perfect control.

L270 Reorientation is not measured at any point, despite the broad assumption within the field parasites can theoreticall apically attach without reorientation, there is also no evidence to say that echinocytosis cannot happen without reorientation. I think it is important not to assume this step has taken place, so I suggest mention of reorientation is removed to stick to directly observed phenotypes.

L296 - I am confused by what 3D and 3E are showing. time from contact to echinocytosis and time of echinocytosis? What period is E measuring? If this is simply from start of the videos then you would instead conclude that the RON11- parasites take longer to make initial contact? Or is it the duration of the echinocytosis itself?

Figure 3 legend. Colours for F do not match between figure and legend (RAP1 is listed as red in legend, but yellow in figure etc).

L440 "supplementation after 46 hours unable to rescue growth"? Should this not be 48 hours onwards? Could the authors clarify the lifecycle length of the parasites they are using? Most lines have a lifecycle length significantly shorter than 48 hours. It may be more logical to talk about hours prior to egress than post invasion in this context.

L584 - Can authors note whether the EF hands are likely in the luminal or cytosolic face.

Minor

L190 remove "As" - there is no follow-on statement within the sentence. Could also change showed to "have shown".

All the best,

 Rob Moon

— — —

ADDITIONAL ACADEMIC EDITOR COMMENTS.

This well written manuscript’s dissection of the time of the invasion defect post attachment, reorientation and deformation of the red blood cell and the demonstration that RON11 is not required for rhoptry secretion are important contributions to understanding the role of RON11 in rhoptries in invasion. The ability to produce only a single rhoptry is also a novel Discovery and important contribution to understanding rhoptry biogenesis.

More detail could be provided for the methods, fig legend and description of the uexm. The authors use the shorthand of Liffner et al of “NHS-ester” for NHS-ester conjugated to a fluorescent dye that non-specifically labels proteins, this needs to be explained by the authors as it is unclear from the current MS how the use of NHS-ester stains the centriolar plaques. The evidence that the knockdown merozoites have only a single rhoptry seems clear and the expanded microscopy clearly indicates segregation of a single rhoptry but it isn’t clear to me that this is occurring after the final mitosis in the knockouts. The basal complex rings are clearly evident in the control in fig 7d but not so clear in the knockout. I agree that the authors explanation is the most likely explanation but it would be strengthened by provision of additional images as a supplement and some analysis of number of segregated single rhoptry vs double in KO and control parasites with a clear basal complex ring would also strengthen the argument.

---

## [Editor Report · Decision Letter 2]

17 Jun 2024

Dear Dr Muralidharan,

Thank you for your patience while we considered your revised manuscript "Plasmodium RON11 triggers biogenesis of the merozoite rhoptry pair and is essential for erythrocyte invasion" for consideration as a Research Article at PLOS Biology. Your revised study has now been evaluated by the PLOS Biology editors, and the Academic Editor. 

In this case, the Academic Editor has arbitrated the revision and agreed that most reviewers’ concerns were addressed. However, the Academic Editor has still some requests which you can find at the end of this e-mail. Specifically, we require that you do the requested non-immune Ig control for the invasion assays. Addressing the concerns and suggestions of the Academic Editor is essential for further consideration of your manuscript for publication in PLOS Biology. Additionally please address the following editorial requirements:

Please supply the numerical values either in the a supplementary file or as a permanent DOI’d deposition for the following figures:

Figure 1DF, 2BC, 3BCDEG, 4B, 5CD, 6, S3ABCDFG, S4BC, S5B

b) Please cite the location of the data clearly in all relevant main and supplementary Figure legends, e.g. “The data underlying this Figure can be found in S1 Data” or “The data underlying this Figure can be found in https://doi.org/10.5281/zenodo.XXXXX”

c) We require the original, uncropped and minimally adjusted images supporting all blot and gel results reported in the following Figures:

Figure 1BC, 5B, S1A, S2ABCD, S5B

We will require these files before a manuscript can be accepted so please prepare and upload them now. Please carefully read our guidelines for how to prepare and upload this data: https://journals.plos.org/plosbiology/s/figures#loc-blot-and-gel-reporting-requirements

d) Please ensure that your Data Statement in the submission system accurately describes where your data can be found and is in final format, as it will be published as written there.

e) Per journal policy, if you have generated any custom code during the curse of this investigation, please make it available without restrictions upon publication. Please ensure that the code is sufficiently well documented and reusable, and that your Data Statement in the Editorial Manager submission system accurately describes where your code can be found.

**IMPORTANT - SUBMITTING YOUR REVISION**

*Resubmission Checklist*

*Published Peer Review*

*PLOS Data Policy*

*Blot and Gel Data Policy*

Sincerely,

Melissa

Melissa Vazquez Hernandez, Ph.D.

Associate Editor

PLOS Biology

ADDITIONAL COMMENTS FROM THE ACADEMIC EDITOR:

The authors have addressed the editor’s comments satisfactorily. They have addressed most of the reviewers’ comments with two exceptions, they have not addressed R2’s request to confirm membrane topology by enzymatic digestion as they say it is beyond the scope of the project. However R2 points out that they claim the N terminus of ron11 is an antibody target during invasion. R2 only says it would be nice to include these data so I think it is not essential, but I think they should remove any speculation about exposure to antibodies if they will not do the digestion assays.

R3 asked for a non-specific antibody control for the growth inhibition assay in fig2c. The authors have not provided this data, the control provided doesn’t address the reviewers request, it only shows that the absorbable anti-rMBP portion of the IgG doesn’t inhibit growth, it does not show that the rat IgG does not have inhibitory effect. This is an easy experiment to do and the authors should have done it.

R2 also asked for the number of rings per schizont rupture to be quantified, the authors said they were unsure how to do this, their fig 2C shows that there are far fewer rings per schizont in the - RON11 than the + RON11. Their invasion assay uses percoll purified schizonts from sorbitol synchronised cultures. I think the reviewer might have been asking for magnet/percoll purified trophozoites to be invasion blocked with heparin, this generates very synchronous late stage schizonts which invade upon removal of the heparin block. This would probably be more accurate but I think the obvious phenotype in the presented data is adequate.

The authors have removed the previous fig S5 so what was s6 is now s5 and they have added a new fig s6 which addresses R2’s request for rhoptry morphology upon KD rescue. This response is sufficient but was difficult to parse, in general the authors wrote a very clear MS but their response did not make clear whether they were reiterating results that were in the original MS, or had added new data, or were just responding to comments without altering the MS. It would be a great aid to assessing their revision if they explicitly indicate exactly what they have changed and where exactly the changes are located.

---

## [Editor Report · Decision Letter 3]

13 Aug 2024

Dear Dr Muralidharan,

Thank you for the submission of your revised Research Article "Plasmodium RON11 triggers biogenesis of the merozoite rhoptry pair and is essential for erythrocyte invasion" for publication in PLOS Biology. On behalf of my colleagues and the Academic Editor, Michael Duffy, I am pleased to say that we can in principle accept your manuscript for publication, provided you address any remaining formatting and reporting issues. These will be detailed in an email you should receive within 2-3 business days from our colleagues in the journal operations team; no action is required from you until then. Please note that we will not be able to formally accept your manuscript and schedule it for publication until you have completed any requested changes.

IMPORTANT: Thank you for supplying the numerical values for the requested figures. However, please cite the location of the data clearly in all relevant main and supplementary Figure legends, e.g. “The data underlying this Figure can be found in S1 Data”. I have asked my colleagues to include this request alongside their own.

PRESS

Sincerely, 

Melissa

Melissa Vazquez Hernandez, Ph.D., Ph.D.

Associate Editor

PLOS Biology
